# Nonconvex Federated Learning on Compact Smooth Submanifolds With Heterogeneous Data

**Jiaojiao Zhang**[1], **Jiang Hu**[2][*], **Anthony Man-Cho So**[3], **Mikael Johansson**[1]

[1] KTH Royal Institute of Technology
[2] University of California, Berkeley     [3] The Chinese University of Hong Kong

{jiaoz, mikaelj}@kth.se, hujiangopt@gmail.com, manchoso@se.cuhk.edu.hk

## Abstract

Many machine learning tasks, such as principal component analysis and low-rank matrix completion, give rise to manifold optimization problems. Although there is a large body of work studying the design and analysis of algorithms for manifold optimization in the centralized setting, there are currently very few works addressing the federated setting. In this paper, we consider nonconvex federated learning over a compact smooth submanifold in the setting of heterogeneous client data. We propose an algorithm that leverages stochastic Riemannian gradients and a manifold projection operator to improve computational efficiency, uses local updates to improve communication efficiency, and avoids client drift. Theoretically, we show that our proposed algorithm converges sub-linearly to a neighborhood of a first-order optimal solution by using a novel analysis that jointly exploits the manifold structure and properties of the loss functions. Numerical experiments demonstrate that our algorithm has significantly smaller computational and communication overhead than existing methods.

## 1 Introduction

Federated learning (FL), which enables clients to collaboratively train models without exchanging their raw data, has gained significant traction in machine learning [1, 2]. The framework is appreciated for its capacity to leverage distributed data, accelerate the training process via parallel computation, and bolster privacy protection. The majority of existing FL algorithms address problems that are either unconstrained or have convex constraints. However, for applications such as principal component analysis (PCA) and matrix completion, where model parameters are subject to nonconvex manifold constraints, there are very few options in the federated setting.

In this paper, we study FL problems over manifolds in the form of

$$\underset{x \in \mathcal{M} \subset \mathbb{R}^{d \times k}}{\text{minimize}} \ f(x) := \frac{1}{n} \sum_{i=1}^{n} f_i(x), \quad f_i(x) = \frac{1}{m_i} \sum_{l=1}^{m_i} f_{il}(x; \mathcal{D}_{il}). \tag{1}$$

Here, $n$ is the number of clients, $x$ is the matrix of model parameters, and $\mathcal{M}$ is a compact smooth submanifold embedded in $\mathbb{R}^{d \times k}$. Examples of such manifolds include the Stiefel, oblique, and symplectic manifolds [3, 4, 5]. For instance, PCA-related optimization problems use the Stiefel manifold $\mathcal{M} = \text{St}(d, k) = \{x \in \mathbb{R}^{d \times k} : x^T x = I_k\}$ to maintain orthogonality [6, 7]. In (1), the global loss $f : \mathbb{R}^{d \times k} \to \mathbb{R}$ is smooth but nonconvex[1], and the local loss function $f_i$ of each client $i$ is the average of the losses $f_{il}$ on the $m_i$ data points in its local dataset $\mathcal{D}_i = \{\mathcal{D}_{i1}, \dots, \mathcal{D}_{im_i}\}$. We consider a heterogeneous data scenario where the statistical properties of $\mathcal{D}_i$ differ across clients.

---

[*]Corresponding author: Jiang Hu
[1]Throughout this paper, all convexity-related concepts are defined in the Euclidean space.

38th Conference on Neural Information Processing Systems (NeurIPS 2024).

Manifold optimization problems of the form (1) appear in many important machine learning tasks, such as PCA [8, 9], low-rank matrix completion [10, 11], multitask learning [12, 13], and deep neural network training [14, 15]. Still, there are very few federated algorithms for machine learning on manifolds. In fact, the work [16] appears to be the only FL algorithm that can deal with manifold optimization problems of a similar generality as ours. Handling manifold constraints in an FL setting poses significant challenges: **(i)** Existing single-machine methods for manifold optimization [6, 5, 4] cannot be directly adapted to the federated setting. Due to the distributed framework, the server has to average the clients local models. Even if each of these models lies on the manifold, their average typically does not due to the nonconvexity of $\mathcal{M}$. The current literature relies on complicated geometric operators like the exponential map, inverse exponential map, and parallel transport, to design an averaging operator for the manifold [16]. However, these mappings may lack closed-form expressions and can be computationally expensive to evaluate. For example, computing the inverse exponential map on the Stiefel manifold requires solving a nonlinear matrix equation [17]. **(ii)** Extending typical FL algorithms to scenarios with manifold constraints is not straightforward, either. Most existing FL algorithms either are unconstrained [18, 19] or only allow for convex constraints [20, 21, 22, 23, 24], but manifold constraints are typically nonconvex. Moreover, compared to nonconvex optimization in Euclidean space, manifold optimization necessitates the consideration of the geometric structure of the manifold and properties of the loss functions, which poses challenges for algorithm design and analysis. **(iii)** Traditional methods for enhancing communication efficiency in FL, like local updates [25], need substantial modifications to accommodate manifold constraints. The so-called client drift issue due to local updates and heterogeneous data [19] persists in the realm of manifold optimization. Directly using client-drift correcting techniques originally developed for Euclidean spaces [19, 26, 27] could lead to additional communication or computational costs due to the manifold constraints. For instance, in [16], the correction term requires additional communication of local Riemannian gradients and involves using parallel transport to move the correction term onto some tangent space in preparation for the exponential mapping. Although some existing decentralized manifold optimization algorithms [9, 28, 29] can be simplified to an FL scenario with only one local update under the assumption of a fully connected network, these algorithms cannot be directly applied to FL scenarios with more than one local update, especially in cases of data heterogeneity. Extending the analysis of these algorithms to FL scenarios with multiple local updates is not straightforward. On the other hand, the use of local updates in FL, compared to these decentralized distributed algorithms, can more effectively reduce the number of communication rounds.

## 1.1 Contributions

We consider the nonconvex FL problem (1) with $\mathcal{M}$ being a compact smooth submanifold and allow for heterogeneous data distribution among clients. Our contributions are summarized as follows.

**1)** We propose an FL algorithm for solving (1) that is efficient in terms of both computation and communication. We employ stochastic Riemannian gradients and a projection operator to address manifold constraints, use local updates to reduce the communication frequency between clients and the server, and design correction terms to overcome client drift. In terms of server updates, our algorithm ensures feasibility of all global model iterates and is computationally efficient since it avoids the techniques used in [16] based on the exponential mapping and inverse exponential mapping for averaging local models on manifolds. For local updates, our algorithm constructs the correction terms locally without increasing communication costs. In comparison, the approach presented in [16] requires each client to transmit an additional local stochastic Riemannian gradient for constructing correction terms. Moreover, [16] necessitates parallel transport to position the correction terms on tangent spaces so that the exponential map can be applied to ensure the feasibility of local models, thereby increasing computational costs. In contrast, our algorithm utilizes a simple projection operator, effectively eliminating the need for parallel transport of correction terms.

**2)** Theoretically, we establish sub-linear convergence to a neighborhood of a first-order optimal solution and demonstrate how this neighborhood depends on the stochastic sampling variance and algorithm parameters. Our analysis introduces novel proof techniques that utilize the curvature of the manifolds and the properties of the loss functions to overcome the challenges posed by the nonconvexity of manifold constraints in the nonconvex FL scenario. Compared to the existing work [16] where analytical results are limited to cases where either the number of local updates is one or the number of participating clients per communication round is one, our theoretical results allow for an arbitrary number of local updates and support full client participation. The key components of our

analysis are the manifold geometry and the Lipschitz continuity of the projection operator, both of which are inherent to the submanifold constraint.

**3)** Our algorithm demonstrates superior performance over alternative methods in the numerical experiments. In particular, it produces high-accuracy results for kPCA and low-rank matrix completion at a significantly lower communication and computation cost than alternative algorithms.

## 1.2 Related work

In this section, we first review federated learning algorithms for composite optimization with and without constraints. Then, we discuss FL algorithms with manifold constraints.

**Composite FL in Euclidean space.** Problem (1) can be viewed as a special case of composite FL where the loss function is a composition of $f$ and the indicator function of $\mathcal{M}$. It is important to note that since the manifold is nonconvex, its indicator function is also nonconvex. Most existing composite FL methods can only handle convex constraints. The work [20] proposed a federated dual averaging method and established its convergence for a general loss function under bounded gradient assumptions, but only for quadratic losses under the bounded heterogeneity assumption that the degree of data heterogeneity among clients is bounded. In contrast, we make no assumptions about the similarity of data across clients. The fast federated dual averaging algorithm [21] extends the work in [20] by using both past gradient information and past model information in the local updates. However, the work [21] requires each client to transmit the local gradient as well as the local model, and it assumes bounded data heterogeneity. The work [22] introduces the federated Douglas-Rachford method, and the work [23] applies this algorithm to solve dual problems. Although these two methods avoid bounded data heterogeneity, they require an increasing number of local updates to ensure convergence, which reduces their practicality in FL. The recent work [24] proposes a communication-efficient FL algorithm that overcomes client drift by decoupling the proximal operator evaluation and the communication and shows that the method converges without any assumptions on data similarity.

**Federated learning on manifolds.** The existing composite FL in Euclidean space [20]-[24] only considers scenarios where the nonsmooth term in the loss functions is convex. However, incorporating a nonconvex manifold constraint as an indicator function introduces a nonconvex nonsmooth term. Consequently, the methods in [20]-[24] are not directly applicable. A typical challenge caused by the nonconvex manifold constraint is that the average of local models, each of which lies on the manifold, may not belong to the manifold. To address this issue, the work [16] introduced Riemannian federated SVRG (RFedSVRG), where the server maps the local models onto a tangent space, calculates an average, and then retracts the average back to the manifold. This process sequentially employs inverse exponential and exponential mappings. Moreover, RFedSVRG employs a correction term to overcome client drift but requires additional communication of local Riemannian gradients to construct the correction term. In addition, the method uses parallel transport to position the correction term, which increases the computation cost even further. Note that the manifold we consider is a compact smooth submanifold embedded in $\mathbb{R}^{d \times k}$, which is more restrictive than the manifolds discussed in [16]. However, this approach still encompasses many common manifolds, including the Stiefel, oblique, and symplectic manifolds [3, 4, 5]. The work [30] explores the differential privacy of RFedSVRG. The work [31] considers the specific manifold optimization problem that appears in PCA and investigates an ADMM-type method that penalizes the orthogonality constraint. However, this algorithm requires solving a subproblem to desired accuracy, which increases computational cost. The work [32] introduces a differentially private FL algorithm for solving PCA. Finally, the works [33] and [34] consider the leading eigenvector problem with homogeneous data across clients, where the loss functions are quadratic and the manifold is a sphere. In contrast, we consider a more general setting with heterogeneous data, where $x$ lies on $\mathcal{M}$ and the loss functions are smooth and nonconvex.

**Notations.** We use $I_k$ to denote a $k \times k$ identity matrix. We use $\| \cdot \|$ to denote Frobenius norm and $\mathrm{tr}(\cdot)$ to denote the trace of a matrix. For a set $\mathcal{B}$, we use $|\mathcal{B}|$ to denote the cardinality. For a random variable $v$, we use $\mathbb{E}[v]$ to denote the expectation and $\mathbb{E}[v|\mathcal{F}]$ to denote the expectation given event $\mathcal{F}$. For an integer $n$, we use $[n]$ to denote the set $\{1, \ldots, n\}$. For two matrices $x, y \in \mathbb{R}^{d \times k}$, we define their Euclidean inner product as $\langle x, y \rangle := \sum_{i=1}^{d} \sum_{j=1}^{k} x_{ij} y_{ij}$. For matrices $z_1, \ldots, z_n \in \mathbb{R}^{d \times k}$, we use $\mathbf{z} := \mathrm{col}\{z_i\}_{i=1}^{n} := [z_1; \ldots; z_n] \in \mathbb{R}^{nd \times k}$ to denote the vertical stack of all matrices. The bold notations $\widehat{\mathbf{z}}$, $\mathbf{c}$, and $\mathbf{\Lambda}$ are defined similarly. Specifically, for a matrix

$x \in \mathbb{R}^{d \times k}$, we define $\mathbf{x} := \mathrm{col}\{x\}_{i=1}^{n} := [x; \ldots; x] \in \mathbb{R}^{nd \times k}$. We use $r$ to denote the index of the communication round and $t$ to denote the index of local updates. Given the local Riemannian gradient $\mathrm{grad} f_i(z_{i,t}^r; \mathcal{B}_{i,t}^r)$ at point $z_{i,t}^r$ with the mini-batch dataset $\mathcal{B}_{i,t}^r$, we define the stack of Riemannian gradients as $\mathrm{grad}\mathbf{f}(\mathbf{z}_t^r; \mathcal{B}_t^r) := \mathrm{col}\{\mathrm{grad} f_i(z_{i,t}^r; \mathcal{B}_{i,t}^r)\}_{i=1}^{n}$ and the stack of average local Riemannian gradients as $\overline{\mathrm{grad}\mathbf{f}}(\mathbf{z}_t^r; \mathcal{B}_t^r) := \mathrm{col}\left\{\frac{1}{n}\sum_{i=1}^{n} \mathrm{grad} f_i(z_{i,t}^r; \mathcal{B}_{i,t}^r)\right\}_{i=1}^{n}$. Given $\mathrm{col}\{z_i\}_{i=1}^{n}$ and $\mathcal{P}_{\mathcal{M}}(z_i)$, we define $\mathcal{P}_{\mathcal{M}}(\mathrm{col}\{z_i\}_{i=1}^{n}) = \mathrm{col}\{\mathcal{P}_{\mathcal{M}}(z_i)\}_{i=1}^{n}$. Given a positive definite matrix $x$, we use $x^{-1/2}$ to denote the inverse of the square root of $x$, i.e., $x^{-1/2}x^{-1/2} = x^{-1}$. We define $\mathcal{D}^2$ to be the second-order differential operator.

## 2 Preliminaries

Below, we introduce fundamental definitions and inequalities for optimization on manifolds.

### 2.1 Optimization on manifolds

Manifold optimization aims to minimize a real-valued function over a manifold, i.e., $\min_{x \in \mathcal{M}} f(x)$. Throughout the paper, we restrict our discussion to embedded submanifolds of the Euclidean space, where the associated topology coincides with the subspace topology of the Euclidean space. We refer to these as embedded submanifolds. Some examples of such manifolds include the Stiefel manifold, oblique manifold, and symplectic manifold [3, 4, 5]. We define the tangent space of $\mathcal{M}$ at point $x$ as $T_x\mathcal{M}$, which contains all tangent vectors to $\mathcal{M}$ at $x$, and the normal space as $N_x\mathcal{M}$ which is orthogonal to the tangent space. With the definition of tangent space, we can define the Riemannian gradient that plays a central role in the characterization of optimality conditions and algorithm design for manifold optimization.

**Definition 2.1** (Riemannian gradient $\mathrm{grad} f(x)$)**.** *The Riemannian gradient $\mathrm{grad} f(x)$ of a function $f$ at the point $x \in \mathcal{M}$ is the unique tangent vector that satisfies*

$$\langle \mathrm{grad} f(x), \xi \rangle_x = df(x)[\xi], \ \forall \xi \in T_x\mathcal{M},$$

*where $\langle \cdot, \cdot \rangle_x$ is the Riemannian metric and $df$ denotes the differential of function $f$.*

For a submanifold $\mathcal{M}$, the Riemannian gradient $\mathrm{grad} f(x)$ (under the Euclidean inner product) can be computed as [5, Proposition 3.61]

$$\mathrm{grad} f(x) = \mathcal{P}_{T_x\mathcal{M}}(\nabla f(x)),$$

where $\mathcal{P}_{T_x\mathcal{M}}(\nabla f(x))$ represents the orthogonal projection of $\nabla f(x)$ onto $T_x\mathcal{M}$. The Riemannian gradient $\mathrm{grad} f(x)$ reduces to the Euclidean gradient $\nabla f(x)$ when $\mathcal{M}$ is the Euclidean space $\mathbb{R}^{d \times k}$.

### 2.2 Proximal smoothness of $\mathcal{M}$

In our federated manifold learning algorithm, the server needs to fuse models that have undergone multiple rounds of local updates by the clients. Due to the nonconvexity of the manifold, the average of points on the manifold is not guaranteed to belong to the manifold. The tangent space-based exponential mapping or other retraction operations commonly used in manifold optimization are expensive in FL [16]. Specifically, the server needs to map the local models onto a tangent space using inverse exponential mapping, calculate an average on the tangent space, and then perform an exponential mapping to retract this average back onto the manifold. This exponential mapping, due to its dependency on the tangent space, also calls for parallel transport during the local updates when there are correction terms. To overcome this difficulty, we use a projection operator $\mathcal{P}_{\mathcal{M}}$ defined by

$$\mathcal{P}_{\mathcal{M}}(x) \in \underset{u \in \mathcal{M}}{\arg\min} \|x - u\|^2 \tag{2}$$

to ensure the feasibility of manifold constraints. The projection operator $\mathcal{P}_{\mathcal{M}}$ can be explicitly calculated for many common submanifolds, as discussed in [35]. For the Stiefel manifold, the closed-form expression for $\mathcal{P}_{\mathcal{M}}(x)$ of a given matrix $x$ with full column rank is $\mathcal{P}_{\mathcal{M}}(x) = x(x^T x)^{-1/2}$; see [35, Proposition 7]. It is worth noting that $\mathcal{P}_{\mathcal{M}}$ can be regarded as a special retraction operator when restricted to the tangent space [35]. However, unlike a typical retraction operator, its domain is $\mathbb{R}^{d \times k}$, not just the tangent space, which enables a more practical averaging operation across clients in

FL. Despite these advantageous properties, the nonconvex nature of $\mathcal{M}$ means that $\mathcal{P}_{\mathcal{M}}(x)$ may be set-valued and non-Lipschitz, making the use and analysis of $\mathcal{P}_{\mathcal{M}}$ in the FL setting highly nontrivial. To tackle this, we introduce the concept of proximal smoothness that refers to a property of a closed set, including $\mathcal{M}$, where the projection becomes a singleton when the point is sufficiently close to the set.

**Definition 2.2** ($\hat{\gamma}$-proximal smoothness of $\mathcal{M}$). *For any $\hat{\gamma} > 0$, we define the $\hat{\gamma}$-tube around $\mathcal{M}$ as*

$$U_{\mathcal{M}}(\hat{\gamma}) := \{x : \mathrm{dist}(x, \mathcal{M}) < \hat{\gamma}\},$$

*where* $\mathrm{dist}(x, \mathcal{M}) := \min_{u \in \mathcal{M}} \|u - x\|$ *is the Eulidean distance between $x$ and $\mathcal{M}$. We say that $\mathcal{M}$ is $\hat{\gamma}$-proximally smooth if the projection operator $\mathcal{P}_{\mathcal{M}}(x)$ is a singleton whenever $x \in U_{\mathcal{M}}(\hat{\gamma})$.*

It is worth noting that any compact smooth submanifold $\mathcal{M}$ embedded in $\mathbb{R}^{d \times k}$ is a proximally smooth set [36, 37]. The constant $\hat{\gamma}$ can be calculated with the method of supporting principle for proximally smooth sets [38, 39]. For instance, the Stiefel manifold is 1-proximally smooth.

**Assumption 2.3.** *The manifold $\mathcal{M}$ is assumed to be a compact smooth submanifold embedded in $\mathbb{R}^{d \times k}$, with the Euclidean inner product serving as its Riemannian metric. Moreover, we assume that the proximal smoothness constant of $\mathcal{M}$ is $2\gamma$.*

With Assumption 2.3, we can ensure not only the uniqueness of the projection but also the Lipschitz continuity of the projection operator $\mathcal{P}_{\mathcal{M}}$ around $\mathcal{M}$, analogous to the non-expansiveness of projections under Euclidean convex constraints.

**Lipschitz continuity of** $\mathcal{P}_{\mathcal{M}}$**.** Define $\overline{U}_{\mathcal{M}}(\gamma) := \{x : \mathrm{dist}(x, \mathcal{M}) \leq \gamma\}$ as the closure of $U_{\mathcal{M}}(\gamma)$. Following the proof in [36, Theorem 4.8], for a $2\gamma$-proximally smooth $\mathcal{M}$, the projection operator $\mathcal{P}_{\mathcal{M}}$ is 2-Lipschitz continuous over $\overline{U}_{\mathcal{M}}(\gamma)$ such that

$$\|\mathcal{P}_{\mathcal{M}}(x) - \mathcal{P}_{\mathcal{M}}(y)\| \leq 2\|x - y\|, \quad \forall x, y \in \overline{U}_{\mathcal{M}}(\gamma). \tag{3}$$

**Normal inequality.** In the normal space $N_x \mathcal{M}$, we exploit the so-called normal inequality [36, 37]. Following [36], given a $2\gamma$-proximally smooth $\mathcal{M}$, for any $x \in \mathcal{M}$ and $v \in N_x \mathcal{M}$, it holds that

$$\langle v, y - x \rangle \leq \frac{\|v\|}{4\gamma} \|y - x\|^2, \quad \forall y \in \mathcal{M}. \tag{4}$$

Intuitively, when $x$ and $y$ are close enough, the matrix $y - x$ is approximately in the tangent space, thus being nearly orthogonal to the normal space.

# 3 Proposed algorithm

In this section, we develop a novel algorithm for nonconvex federated learning on manifolds. The algorithm is inspired by the proximal FL algorithm for strongly convex problems in Euclidean space recently proposed in [24] but includes several non-trivial extensions. These include the use of Riemannian gradients and manifold projection operators and the ability to handle nonconvex loss functions, which call for a different convergence analysis.

## 3.1 Algorithm description

The per-client implementation of our algorithm is detailed in Algorithm 1. Similarly to the well-known FedAvg, it operates in a federated learning setting with one server and $n$ clients. Each client $i$ engages in $\tau$ steps of local updates before updating the server. We use $r$ as the index of communication rounds and $t$ as the index of local updates.

At any communication round $r$, client $i$ downloads the global model $x^r$ from the server and computes $\mathcal{P}_{\mathcal{M}}(x^r)$. Each client $i$ updates two local variables, $\hat{z}_{i,t}^r$ and $z_{i,t}^r$, where $\hat{z}_{i,t}^r$ aggregates the Riemannian gradients from local updates, and $z_{i,t}^r = \mathcal{P}_{\mathcal{M}}(\hat{z}_{i,t}^r)$ ensures that Riemannian gradients can be computed at points on $\mathcal{M}$. The update of $\hat{z}_{i,t}^r$ is given in Line 8, where $\mathcal{B}_{i,t}^r$ is a mini-batch dataset and $c_i^r$ is a correction term to eliminate client drift. After $\tau$ local updates, client $i$ sends $\hat{z}_{i,\tau}^r$ to the server.

The server receives all $\hat{z}_{i,\tau}^r$, computes their average to form the global model $x^{r+1}$ following Line 13, and broadcasts $x^{r+1}$ to each client $i$ that uses $x^{r+1}$ to locally construct the correction term $c_i^{r+1}$.

---

**Algorithm 1** Proposed algorithm

---

1: **Input:** $R, \tau, \eta, \eta_g, \tilde{\eta} = \eta\eta_g\tau, x^1$, and $c_i^1 = 0$ for all $i \in [n]$
2: **for** $r = 1, 2, \ldots, R$ **do**
3:     **Client** $i$
4:     Set $\widehat{z}_{i,0}^r = \mathcal{P}_\mathcal{M}(x^r)$ and $z_{i,0}^r = \mathcal{P}_\mathcal{M}(x^r)$
5:     **for** $t = 0, 1, \ldots, \tau - 1$ **do**
6:         Sample a mini-batch dataset $\mathcal{B}_{i,t}^r \subseteq \mathcal{D}_i$ with $|\mathcal{B}_{i,t}^r| = b$
7:         Update $\mathrm{grad}f_i(z_{i,t}^r; \mathcal{B}_{i,t}^r) = \frac{1}{b}\sum_{\mathcal{D}_{il}\in\mathcal{B}_{i,t}^r} \mathrm{grad}f_{il}(z_{i,t}^r; \mathcal{D}_{il})$
8:         Update $\widehat{z}_{i,t+1}^r = \widehat{z}_{i,t}^r - \eta\left(\mathrm{grad}f_i(z_{i,t}^r; \mathcal{B}_{i,t}^r) + c_i^r\right)$
9:         Update $z_{i,t+1}^r = \mathcal{P}_\mathcal{M}\left(\widehat{z}_{i,t+1}^r\right)$
10:     **end for**
11:     Send $\widehat{z}_{i,\tau}^r$ to the server
12:     **Server**
13:     Update $x^{r+1} = \mathcal{P}_\mathcal{M}(x^r) + \eta_g\left(\frac{1}{n}\sum_{i=1}^n \widehat{z}_{i,\tau}^r - \mathcal{P}_\mathcal{M}(x^r)\right)$
14:     Broadcast $x^{r+1}$ to all the clients
15:     **Client** $i$
16:     Receive $x^{r+1}$
17:     Update $c_i^{r+1} = \frac{1}{\eta_g\eta\tau}(\mathcal{P}_\mathcal{M}(x^r) - x^{r+1}) - \frac{1}{\tau}\sum_{t=0}^{\tau-1} \mathrm{grad}f_i(z_{i,t}^r; \mathcal{B}_{i,t}^r)$
18: **end for**
19: **Output:** $\mathcal{P}_\mathcal{M}(x^{R+1})$

---

In the proposed algorithm, each client $i$ downloads $x^r$ at the start of local updates and uploads $\hat{z}_{i,\tau}^r$ at the end of the local updates. Therefore, each communication round involves each client and the server exchanging only a single $d \times k$ matrix.

### 3.2 Algorithm intuition and innovations

To better understand the proposed algorithm, we present its equivalent and more compact form:

$$
\begin{cases}
\widehat{\mathbf{z}}_{t+1}^r = \widehat{\mathbf{z}}_t^r - \eta\Big(\mathrm{grad}\mathbf{f}\left(\mathbf{z}_t^r; \mathcal{B}_t^r\right) + \frac{1}{\tau}\sum_{t=0}^{\tau-1} \overline{\mathrm{grad}\mathbf{f}}\left(\mathbf{z}_t^{r-1}; \mathcal{B}_t^{r-1}\right) - \frac{1}{\tau}\sum_{t=0}^{\tau-1} \mathrm{grad}\mathbf{f}\left(\mathbf{z}_t^{r-1}; \mathcal{B}_t^{r-1}\right)\Big), \\
\mathbf{z}_{t+1}^r = \mathcal{P}_\mathcal{M}\left(\widehat{\mathbf{z}}_{t+1}^r\right), \\
\mathbf{x}^{r+1} = \mathcal{P}_\mathcal{M}(\mathbf{x}^r) - \eta_g\eta\sum_{t=0}^{\tau-1} \overline{\mathrm{grad}\mathbf{f}}\left(\mathbf{z}_t^r; \mathcal{B}_t^r\right).
\end{cases}
\tag{5}
$$

For the initialization of correction term, we set $\mathrm{grad}f_i\left(z_{i,t}^0; \mathcal{B}_{i,t}^0\right) = 0$ for all $t$ and $i$ so that $\frac{1}{\tau}\sum_{t=0}^{\tau-1} \overline{\mathrm{grad}\mathbf{f}}\left(\mathbf{z}_t^0; \mathcal{B}_t^0\right) - \frac{1}{\tau}\sum_{t=0}^{\tau-1} \mathrm{grad}\mathbf{f}\left(\mathbf{z}_t^0; \mathcal{B}_t^0\right) = 0$, which coincides with the initialization $c_i^1 = 0$ in Algorithm 1. The equivalence between Algorithm 1 and (5) can be proved following the similar derivations in [24] and is therefore omitted.

With (5), we highlight the key properties and innovations of the proposed algorithm.

**1) Recovery of the centralized algorithm in special cases.** Substituting the definitions of $\mathbf{x}$, $\overline{\mathrm{grad}\mathbf{f}}\left(\mathbf{z}_t^r; \mathcal{B}_t^r\right)$, and $\tilde{\eta}$ into the last step in (5), we have

$$
\mathcal{P}_\mathcal{M}(x^{r+1}) = \mathcal{P}_\mathcal{M}\Big(\mathcal{P}_\mathcal{M}(x^r) - \tilde{\eta}\frac{1}{n\tau}\sum_{i=1}^n \sum_{t=0}^{\tau-1} \left(\mathrm{grad}f_i\left(z_{i,t}^r; \mathcal{B}_{i,t}^r\right)\right)\Big).
\tag{6}
$$

Thanks to the introduction of the variable $\hat{z}_{i,t}^r$ during the local updates for each client $i$ in Algorithm 1, the server after averaging $\hat{z}_{i,\tau}^r$ obtains an accumulation of $\tau$ local Riemannian gradients across local updates and an average of the local Riemannian gradients across all clients. In the special case where $\tau = 1$ and $b = m_i$, i.e., with the local full Riemannian gradient for each client $i$, the update of (6) recovers the centralized projected Riemannian gradient descent (C-PRGD)

$$
\tilde{x}^{r+1} := \mathcal{P}_\mathcal{M}\left(\mathcal{P}_\mathcal{M}(x^r) - \tilde{\eta}\mathrm{grad}f(\mathcal{P}_\mathcal{M}(x^r))\right).
\tag{7}
$$

In our analysis, we will compare the sequence $\mathcal{P}_{\mathcal{M}}(x^{r+1})$ generated by our algorithm with the virtual iterate $\tilde{x}^{r+1}$ to establish the convergence of our algorithm.

**2) Feasibility of all iterates at a low computational cost.** Our algorithm uses $\mathcal{P}_{\mathcal{M}}$ to obtain feasible solutions on the manifold, which is computationally more efficient than the commonly used exponential mapping. In fact, since the exponential mapping relies on a point on the manifold and the tangent space at that point, it cannot be directly used in our algorithm. In the local updates, it is difficult to perform exponential mapping on $\hat{\mathbf{z}}_{t+1}^r$ because $\hat{\mathbf{z}}_t^r$ is not on the manifold; see the first step in (5). As shown in [24], $\hat{\mathbf{z}}_{t+1}^r$ is essential for the server to obtain aggregated Riemannian gradients from $n$ clients after $\tau$ local updates. Moreover, at the server, although $\mathcal{P}_{\mathcal{M}}(\mathbf{x}^r)$ is on the manifold, the aggregated direction does not lie in the tangent space at $\mathcal{P}_{\mathcal{M}}(\mathbf{x}^r)$. The algorithm suggested in [16] uses an exponential mapping to fuse local models. It needs to map the local models to a tangent space using the inverse exponential mapping and then retract the result back to the manifold, which is computationally expensive. Our use of $\mathcal{P}_{\mathcal{M}}$ on a point in the Euclidean space close to the manifold avoids these high computational costs, but creates new challenges for the analysis.

**3) Overcoming client drift.** Inspired by [24], we use a correction term $c_i^r$ to address client drift. According to the first step of (5), the correction employs the idea of "variance reduction", which involves replacing the old local Riemannian gradient $\frac{1}{\tau}\sum_{t=0}^{\tau-1} \mathrm{grad}\mathbf{f}\left(\mathbf{z}_t^{r-1};\mathcal{B}_t^{r-1}\right)$ with the new one $\mathrm{grad}\mathbf{f}\left(\mathbf{z}_t^r;\mathcal{B}_t^r\right)$ in the average of all client Riemannian gradients $\frac{1}{\tau}\sum_{t=0}^{\tau-1}\overline{\mathrm{grad}\mathbf{f}}\left(\mathbf{z}_t^{r-1};\mathcal{B}_t^{r-1}\right)$, where the "variance" refers to the differences in Riemannian gradients among clients caused by data heterogeneity. Compared to [16], our correction improves communication and computation. The correction approach in [16] necessitates extra transmissions of local Riemannian gradients, while our correction term can be locally generated, leading to a significantly reduced communication overhead. Furthermore, [16] employs parallel transport to position the correction term with a specific tangent space for the exponential mapping to ensure local model feasibility. Our approach, which utilizes $\mathcal{P}_{\mathcal{M}}$, eliminates the need for parallel transport and reduces the computations per iteration even further.

## 4 Analysis

In this section, we analyze the convergence of the proposed Algorithm 1. Throughout the paper, we make the following assumptions, which are common in manifold optimization.

**Assumption 4.1.** *Each $f_{il}(x;\mathcal{D}_{il}) : \mathbb{R}^{d\times k} \mapsto \mathbb{R}$ has $\hat{L}$-Lipschitz continuous gradient $\nabla f_{il}(x;\mathcal{D}_{il})$ on the convex hull of $\mathcal{M}$, denoted by $\mathrm{conv}(\mathcal{M})$, i.e., for any $x,y \in \mathrm{conv}(\mathcal{M})$, it holds that*

$$\|\nabla f_{il}(x;\mathcal{D}_{il}) - \nabla f_{il}(y;\mathcal{D}_{il})\| \leq \hat{L}\|x - y\|. \tag{8}$$

With the compactness of $\mathcal{M}$, there exists a constant $D_f > 0$ such that the Euclidean gradient $\nabla f_{il}(x;\mathcal{D}_{il})$ of $f_{il}$ is bounded by $D_f$, i.e., $\max_{i,l,x\in\mathcal{M}}\|\nabla f_{il}(x;\mathcal{D}_{il})\| \leq D_f$. It then follows from [28, Lemma 4.2] that there exists a constant $\hat{L} \leq L < \infty$ such that for any $x,y \in \mathcal{M}$,

$$f_{il}(y;\mathcal{D}_{il}) \leq f_{il}(x;\mathcal{D}_{il}) + \langle \mathrm{grad}f_{il}(x;\mathcal{D}_{il}), y - x\rangle + \frac{L}{2}\|x - y\|^2,$$

$$\|\mathrm{grad}f_{il}(x;\mathcal{D}_{il}) - \mathrm{grad}f_{il}(y;\mathcal{D}_{il})\| \leq L\|x - y\|.$$

To address the stochasticity introduced by the random sampling $\mathcal{B}_{i,t}^r$, we define $\mathcal{F}_t^r$ as the $\sigma$-algebra generated by the set $\{\mathcal{B}_{i,\tilde{t}}^{\tilde{r}} \mid i \in [n], \tilde{r} \in [r], \tilde{t} \in [t-1]\}$ and make the following assumptions regarding the stochastic Riemannian gradients, similar to [40, Assumption 2].

**Assumption 4.2.** *Each stochastic Riemannian gradient $\mathrm{grad}f_i(z_{i,t}^r;\mathcal{B}_{i,t}^r)$ in Algorithm 1 satisfies*

$$\mathbb{E}\left[\mathrm{grad}f_i(z_{i,t}^r;\mathcal{B}_{i,t}^r)|\mathcal{F}_t^r\right] = \mathrm{grad}f_i(z_{i,t}^r),$$

$$\mathbb{E}\left[\left\|\mathrm{grad}f_i(z_{i,t}^r;\mathcal{B}_{i,t}^r) - \mathrm{grad}f_i(z_{i,t}^r)\right\|^2|\mathcal{F}_t^r\right] \leq \frac{\sigma^2}{b}. \tag{9}$$

Considering the nonconvexity of $f$ and the manifold constraints, we characterize the first-order optimality of (1). A point $x^\star$ is defined as a first-order optimal solution of (1) if $x^\star \in \mathcal{M}$ and $\mathrm{grad}f(x^\star) = 0$. We employ the norm of $\mathcal{G}_{\tilde{\eta}}(\mathcal{P}_{\mathcal{M}}(x^r))$ as a suboptimality metric, defined as

$$\mathcal{G}_{\tilde{\eta}}(\mathcal{P}_{\mathcal{M}}(x^r)) := \frac{1}{\tilde{\eta}}(\mathcal{P}_{\mathcal{M}}(x^r) - \tilde{x}^{r+1}), \tag{10}$$

and $\tilde{x}^{r+1}$ defined in (7) is used only for analytical purposes. In optimization on Euclidean space such that $\mathcal{M} = \mathbb{R}^{d \times k}$, the quantity $\mathcal{G}_{\tilde{\eta}}(\mathcal{P}_\mathcal{M}(x^r))$ serves as a widely accepted metric to assess first-order optimality for nonconvex composite problems [41]. In optimization on manifold, we have $\mathcal{G}_{\tilde{\eta}}(\mathcal{P}_\mathcal{M}(x^r)) = 0$ if and only if $\operatorname{grad} f(\mathcal{P}_\mathcal{M}(x^r)) = 0$ for any $\tilde{\eta} > 0$. Moreover, for a suitable $\tilde{\eta}$, the use of Riemannian gradients in the update of $\tilde{x}^{r+1}$ helps us to establish a quasi-isometric property, specifically that $1/2\|\operatorname{grad} f(\mathcal{P}_\mathcal{M}(x^r))\| \leq \|\mathcal{G}_{\tilde{\eta}}(\mathcal{P}_\mathcal{M}(x^r))\| \leq 2\|\operatorname{grad} f(\mathcal{P}_\mathcal{M}(x^r))\|$; see Lemmas A.1 and A.2. With $\mathcal{G}_{\tilde{\eta}}(\mathcal{P}_\mathcal{M}(x^r))$, we have the following theorem.

**Theorem 4.3.** *Under Assumptions 2.3, 4.1, and 4.2, if the step sizes satisfy*

$$\tilde{\eta} := \eta_g \eta \tau \leq \min\left\{ \frac{1}{24ML}, \frac{\gamma}{6D_f}, \frac{1}{D_f L_\mathcal{P}} \right\}, \quad \eta_g = \sqrt{n}, \tag{11}$$

*where* $M = \max\{\operatorname{diam}(\mathcal{M})/\gamma, 2\}$, $\operatorname{diam}(\mathcal{M}) = \max_{x,y \in \mathcal{M}}\|x - y\|$, $D_f = \max_{i,l,x \in \mathcal{M}} \|\nabla f_{il}(x; \mathcal{D}_{il})\|$, *and* $L_\mathcal{P} = \max_{x \in \overline{U}_\mathcal{M}(\gamma)} \|\mathcal{D}^2 \mathcal{P}_\mathcal{M}(x)\|$, *then the sequence* $\mathcal{P}_\mathcal{M}(x^r)$ *generated by Algorithm 1 satisfies*

$$\frac{1}{R} \sum_{r=1}^{R} \mathbb{E}\|\mathcal{G}_{\tilde{\eta}}(\mathcal{P}_\mathcal{M}(x^r))\|^2 \leq \frac{8\Omega^1}{\sqrt{n}\eta\tau R} + \frac{64\sigma^2}{n\tau b}, \tag{12}$$

*where* $\Omega^1 > 0$ *is a constant related to initialization.*

In Theorem 4.3, the first term on the right hand of (12) converges at a sub-linear rate, which is common for constrained nonconvex optimization [41, 42]. The second term is a constant error caused by the variance $\sigma^2$ of stochastic Riemannian gradients.

**Theoretical contributions.** The work [16] establishes convergence rates of $\mathcal{O}(1/R)$ for $\tau = 1$ and $\mathcal{O}(1/(\tau R))$ for $\tau > 1$ but only if a single client participates in the training per communication round. In contrast, our Theorem 4.3 achieves a rate of $\mathcal{O}(1/(\sqrt{n}\tau R))$ for $\tau > 1$ and full client participation. Our theorem indicates that multiple local updates enable faster convergence, which distinguishes our algorithm from decentralized manifold optimization algorithms [9, 28] that limit clients to do a single local update. The works [43] and [1] study smooth nonconvex FL with heterogeneous data in Euclidean space, relying on the assumptions of bounded second moments [43, Assumption 1] and $B$-local dissimilarity [1, Definition 3], respectively, to establish convergence. These assumptions imply some similarity between client data. Our main analytical advantages over [43] and [1] are twofold: We do not assume data similarity, and we completely eliminate client drift. This is reflected in our convergence error, which eliminates the error introduced by data heterogeneity. Our convergence analysis relies on several novel techniques. Specifically, we capitalize on the structure of $\mathcal{M}$ and exploit the proximal smoothness of $\mathcal{M}$ to guarantee the uniqueness of $\mathcal{P}_\mathcal{M}$ and Lipschitz continuity of $\mathcal{P}_\mathcal{M}$ within a tube around $\mathcal{M}$. Additionally, we carefully select the step sizes to ensure that the iterates remain close to $\mathcal{M}$, thus preserving the established properties throughout the iterations. Last but not least, we select an appropriate first-order optimality metric (see (10)) and jointly consider the properties of $\mathcal{M}$ and the loss functions to establish some new inequalities for the convergence of this metric, given in Appendix.

## 5 Numerical experiments

In this section, we conduct numerical experiments on two applications on the Stiefel manifold: kPCA and the low rank matrix completion (LRMC). We compare with existing algorithms, including RFedavg, RFedprox, and RFedSVRG. RFedavg and RFedprox are direct extensions of FedAvg [25] and Fedprox [1]. For RFedSVRG, there are no theoretical guarantees when we set $\tau > 1$ and make all clients participate. In all alternative algorithms, the calculations of the exponential mapping, its inverse, and the parallel transport on the Stiefel manifold are needed. The exponential mapping has a closed-form expression but involves a matrix exponential [6], the inverse exponential mapping needs to solve a nonlinear matrix equation [17], and the parallel transport needs to solve a linear differential equation [44], all of which are computationally challenging. In their implementations, approximate versions of these mappings are used [16, 45, 46].

**kPCA.** Consider the kPCA problem

$$\underset{x \in \operatorname{St}(d,k)}{\operatorname{minimize}} f(x) = \frac{1}{n} \sum_{i=1}^{n} f_i(x), \quad f_i(x) = -\frac{1}{2}\operatorname{tr}(x^T A_i^T A_i x),$$

where $\mathrm{St}(d,k) = \{x \in \mathbb{R}^{d \times k} \mid x^T x = I_k\}$ denotes the Stiefel manifold, and $A_i^T A_i \in \mathbb{R}^{d \times d}$ is the covariance matrix of the local data $A_i \in \mathbb{R}^{p \times d}$ of client $i$. We conduct experiments where the matrix $A_i$ is from the Mnist dataset. The specific experiment settings can be found in Appendix A.4.1.

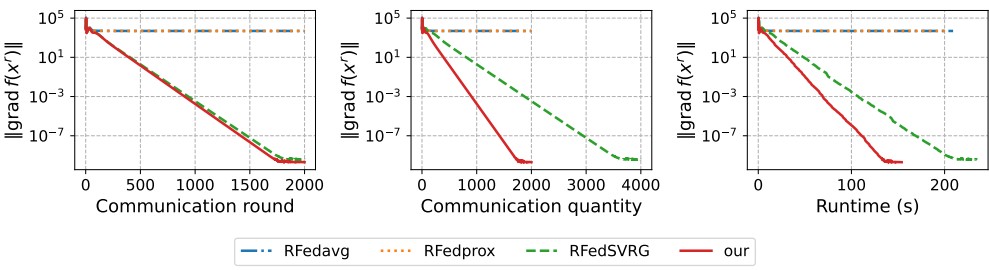

Figure 1: kPCA problem with Mnist dataset: Comparison on $\|\mathrm{grad} f(x^r)\|$.

In the first set of experiments, we compare our algorithm with RFedavg, RFedprox, and RFedSVRG. RFedSVRG requires each client to transmit two $d \times k$ matrices per communication round, while our algorithm only transmits a single matrix. We use communication quantity to count the total number of $d \times k$ matrices that per client transmits to the server. We use the local full gradient $\nabla f_i$ to mitigate the effects of stochastic gradient noise. In Figs. 1, we set $\tau = 10$ and $\eta = 1/\beta$ for all algorithms, where $\beta$ is the square of the largest singular value of $\mathrm{col}\{A_i\}_{i=1}^n$. We set $\eta_g = 1$ to facilitate comparison with other algorithms. As noted below (40) in the Appendix, all analytical results leading up to (40) remain valid for $\eta_g = 1$. It can be observed that RFedavg and RFedprox face the issue of client drift and have low accuracy. Both RFedSVRG and our algorithm overcome the client drift, but our algorithm, though being similar in terms of communication rounds, is much faster in terms of communication quantity and run time.

In the second set of experiments, we test the impact of $\tau$. For all the algorithms, we set the step size $\eta = 1/\beta$ and $\tau \in \{10, 15, 20\}$. For our algorithm, we set $\eta_g = 1$. The experiment results are shown in Figs. 2. For all values of $\tau$, our algorithm achieves better convergence and requires less communication quantity.

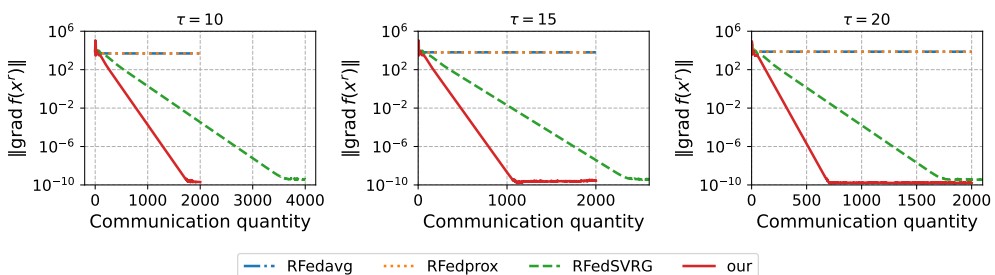

Figure 2: kPCA with Mnist dataset: The impacts of $\tau$.

In addition, we test the impact of stochastic Riemannian gradients with different batch sizes. We set $\eta = 1/(20\beta)$. As shown in Figs. 3, our algorithm converges to a neighborhood due to the sampling noise and larger batch size leads to faster convergence. Additional experimental results can be found in Appendix A.4.1.

**LRMC.** LRMC aims to recover a low-rank matrix $A \in \mathbb{R}^{d \times T}$ from its partial observations. Let $\Omega$ be the set of indices of known entries in $A$, the rank-$k$ LRMC problem can be written as $\mathrm{minimize}_{X \in \mathrm{St}(d,k), V \in \mathbb{R}^{k \times T}} \frac{1}{2} \|\mathcal{P}_\Omega(XV - A)\|^2$, where the projection operator $\mathcal{P}_\Omega$ is defined in an entry-wise manner with $(\mathcal{P}_\Omega(A))_{l_1 l_2} = A_{l_1 l_2}$ if $(l_1, l_2) \in \Omega$ and 0 otherwise. In terms of the FL setting, we consider the case where the observed data matrix $\mathcal{P}_\Omega(A)$ is equally divided into $n$ clients

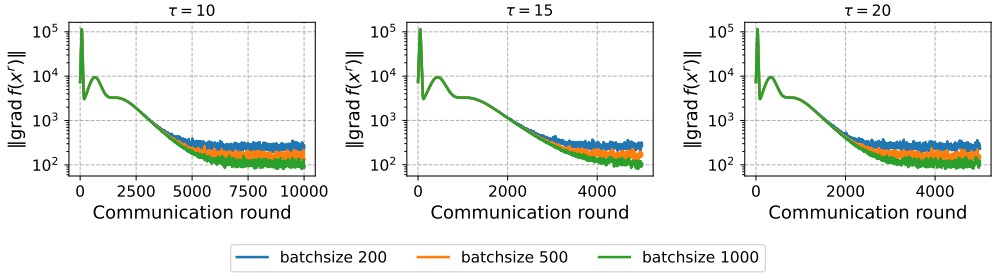

Figure 3: kPCA problem with Mnist dataset: The impacts of stochastic Riemannian gradients.

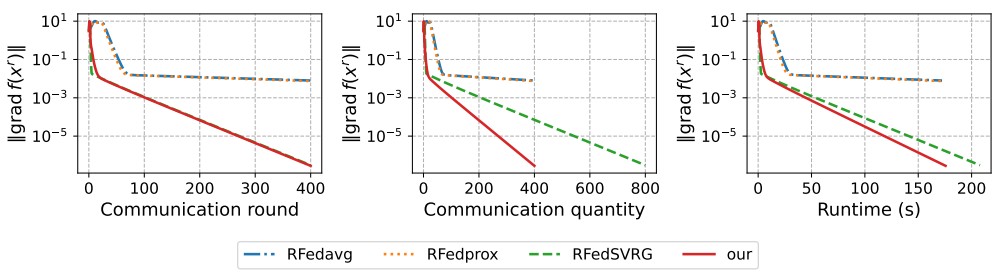

Figure 4: LRMC: Comparison on $\|\mathrm{grad}f(x^r)\|$.

by columns, denoted by $A_1, \ldots, A_n$. Then, the FL LRMC problem is

$$\underset{X \in \mathrm{St}(d,k)}{\text{minimize}} \quad \frac{1}{2n} \sum_{i=1}^{n} \|\mathcal{P}_{\Omega_i}(XV_i(X) - A_i)\|^2, \tag{13}$$

where $\Omega_i$ is the subset corresponding to client $i$ in $\Omega$ and $V_i(X) := \mathrm{argmin}_V \|\mathcal{P}_{\Omega_i}(XV - A_i)\|$. In the experiments, we set $T = 1000$, $d = 100$, $k = 2$, $n = 10$, and use the local full gradients. The other settings can be found in Appendix A.4.2.

The numerical comparisons with RFedavg, RFedprox, and RFedSVRG are presented in Figs. 4. Our algorithm and RFedSVRG achieve similar convergence for communication rounds, but our algorithm converges faster than RFedSVRG in terms of communication quantity and run time. Additional experimental results can be found in Appendix A.4.2.

## 6  Conclusions and limitations

This paper addresses the challenges of FL on compact smooth submanifolds. We introduce a novel algorithm that enables full client participation, local updates, and heterogeneous data distributions. By leveraging stochastic Riemannian gradients and a manifold projection operator, our method enhances computational and communication efficiency while mitigating client drift. By exploiting the manifold structure and properties of the loss function, we prove sub-linear convergence to a neighborhood of a first-order stationary point. Numerical experiments show a superior performance of our algorithm in terms of computational and communication costs compared to the state-of-the-art.

Our paper motivates several questions for further investigation. First, the absence of closed-form solutions for the projection operator $\mathcal{P}_{\mathcal{M}}$ for certain manifolds necessitates exploring methods to calculate projections approximately. Additionally, our step-size selection relies on the proximal smoothness constant $\gamma$, underscoring the need for estimating $\gamma$ either off-line for specific manifolds or adaptively on-line. Furthermore, designing algorithms for partial participation and devising corresponding client-drift correction mechanisms require further investigation.

## Acknowledgments

This research is supported in part by the Hong Kong Research Grant Council (RGC) through the General Research Fund (GRF) project CUHK 14205421, in part by the Knut and Alice Wallenberg Foundation through grant 2022.0050, and in part by the Swedish Research Council through grant 2023-05538.

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

# A  Appendix

## A.1  Notations

We analyze the proposed algorithm using the Lyapunov function $\Omega^r$ defined by

$$\Omega^r := f(\mathcal{P}_{\mathcal{M}}(x^r)) - f^\star + \frac{1}{n\tilde{\eta}}\|\mathbf{\Lambda}^r - \overline{\mathbf{\Lambda}}^r\|^2, \tag{14}$$

where $f^\star$ is the optimal value of problem (1) and we define

$$\mathbf{\Lambda}^r := \eta(\tau \mathrm{grad}\mathbf{f}(\mathcal{P}_{\mathcal{M}}(\mathbf{x}^r)) + \sum_{t=0}^{\tau-1}\overline{\mathrm{grad}\mathbf{f}(\mathbf{z}_t^{r-1};\mathcal{B}_t^{r-1})} - \sum_{t=0}^{\tau-1}\mathrm{grad}\mathbf{f}(\mathbf{z}_t^{r-1};\mathcal{B}_t^{r-1}))$$

and $\overline{\mathbf{\Lambda}}^r := \mathrm{col}\left\{\frac{1}{n}\sum_{i=1}^{n}\Lambda_i^r\right\}_{i=1}^{n}$.

The Lyapunov function consists of two parts: to bound the suboptimality of the global model $\mathcal{P}_{\mathcal{M}}(x^r)$ and the reduction of "variance" among clients, respectively.

## A.2  Preliminary lemmas

Let us start with the following lemma on the global-like Lipschitz-continuity property of $\mathcal{P}_{\mathcal{M}}$.

**Lemma A.1.** *There exists a constant $M > 0$ such that for any $x \in \mathcal{M}$,*

$$\|\mathcal{P}_{\mathcal{M}}(x + u) - x\| \leq M\|u\|. \tag{15}$$

*Proof.* Let us consider two cases:

- $\|u\| \geq \gamma$: Since $\mathcal{P}_{\mathcal{M}}(x + u)$ and $x$ belong to $\mathcal{M}$, we have

$$\|\mathcal{P}_{\mathcal{M}}(x + u) - x\| \leq \mathrm{diam}(\mathcal{M}) \leq \frac{\mathrm{diam}(\mathcal{M})}{\gamma}\|u\|,$$

  where $\mathrm{diam}(\mathcal{M}) := \max_{x,y\in\mathcal{M}}\|x - y\|$ is the diameter of $\mathcal{M}$.

- $\|u\| \leq \gamma$: By the 2-Lipschitz continuity of $\mathcal{P}_{\mathcal{M}}$ over $\overline{U}_{\mathcal{M}}(\gamma)$ in (3), we have

$$\|\mathcal{P}_{\mathcal{M}}(x + u) - x\| \leq 2\|u\|.$$

Setting $M := \max\left\{\frac{\mathrm{diam}(\mathcal{M})}{\gamma}, 2\right\}$, we complete the proof. $\square$

In the following, we show the reasonableness of the suboptimality metric $\|\mathcal{G}_{\tilde{\eta}}(\cdot)\|$.

**Lemma A.2.** *Consider $\mathcal{G}_{\tilde{\eta}}(\cdot)$ defined by (10). Then, for any $x \in \mathcal{M}$, it holds that*

$$\mathrm{grad}f(x) = 0 \quad \text{if and only if} \quad \mathcal{G}_{\tilde{\eta}}(x) = 0.$$

*In addition, under Assumptions 2.3 and 4.1, if $\tilde{\eta} \leq \min\left\{\frac{\gamma}{D_f}, \frac{1}{D_f L_{\mathcal{P}}}\right\}$ with $L_{\mathcal{P}}$ being the smoothness constant of $\mathcal{D}^2\mathcal{P}_{\mathcal{M}}(\cdot)$ over $\overline{U}_{\mathcal{M}}(\gamma)$, it holds that*

$$\|\mathrm{grad}f(x)\| \leq 2\|\mathcal{G}_{\tilde{\eta}}(x)\|. \tag{16}$$

*Proof.* If $\mathrm{grad}f(x) = 0$, it follows directly from the definition of $\mathcal{G}_{\tilde{\eta}}(\cdot)$ that $\mathcal{G}_{\tilde{\eta}}(x) = 0$. Conversely, if $\mathcal{G}_{\tilde{\eta}}(x) = 0$, we have

$$x = \mathcal{P}_{\mathcal{M}}(x - \tilde{\eta}\mathrm{grad}f(x)) := \operatorname*{argmin}_{y\in\mathcal{M}}\|y - x + \tilde{\eta}\mathrm{grad}f(x)\|^2.$$

It follow from the optimality of $x$ that $0 = P_{T_x\mathcal{M}}(\tilde{\eta}\mathrm{grad}f(x))$, which implies that $\mathrm{grad}f(x) = 0$.

With [47, Lemma], $\mathcal{P}_{\mathcal{M}}(\cdot)$ is sufficiently smooth over $\overline{U}_{\mathcal{M}}(\gamma)$. Let us define $L_{\mathcal{P}} := \max_{x \in \overline{U}_{\mathcal{M}}(\gamma)} \|\mathcal{D}^2 \mathcal{P}_{\mathcal{M}}(x)\|$, where $\mathcal{D}^2$ denotes the second-order differential operator. Then, we have

$$\|\mathcal{G}_{\tilde{\eta}}(x)\| = \frac{1}{\tilde{\eta}} \|x - \mathcal{P}_{\mathcal{M}}(x - \tilde{\eta} \mathrm{grad} f(x))\|$$

$$\geq \|\mathrm{grad} f(x)\| - \frac{1}{2} L_{\mathcal{P}} \tilde{\eta} \|\mathrm{grad} f(x)\|^2$$

$$\geq \frac{1}{2} \|\mathrm{grad} f(x)\|,$$

where we use $\tilde{\eta} \leq \frac{\gamma}{D_f}$ in the first inequality and $\tilde{\eta} \leq \frac{1}{L_{\mathcal{P}} D_f}$ in the second inequality. This gives (16). $\qquad \square$

To prove Theorem 4.3, we use the following lemma to establish a recursion on the second term on $\mathbf{\Lambda}^r$ in the Lyapunov function.

**Lemma A.3.** *Under Assumptions 2.3, 4.1, and 4.2, if $\tilde{\eta} \leq \min\left\{ \frac{\eta_g}{16L}, \frac{\gamma \eta_g}{2D_f} \right\}$, we have*

$$\frac{1}{n} \mathbb{E}\|\mathbf{\Lambda}^{r+1} - \overline{\mathbf{\Lambda}}^{r+1}\|^2 - 2\eta^2 \tau^2 L^2 \mathbb{E}\left\|\mathcal{P}_{\mathcal{M}}(x^{r+1}) - \mathcal{P}_{\mathcal{M}}(x^r)\right\|^2 \tag{17}$$

$$\leq \frac{1}{n} 4\eta^2 \tau L^2 \left( 3nM^2 \tau^3 \eta^2 \|\mathrm{grad} f(\mathcal{P}_{\mathcal{M}}(x^r))\|^2 + 9\tau \mathbb{E}\|\mathbf{\Lambda}^r - \overline{\mathbf{\Lambda}}^r\|^2 + 18n\tau^2 \eta^2 \frac{\sigma^2}{b} \right) + \frac{1}{n} 4\eta^2 n^2 \tau^2 \frac{\sigma^2}{n\tau b}.$$

*Proof.* As a first step, we bound the drift error $\left\|z_{i,t+1}^r - \mathcal{P}_{\mathcal{M}}(x^r)\right\|^2$ that is caused by the local updates. If $\tau = 1$, the error is zero since $z_{i,t}^r = \mathcal{P}_{\mathcal{M}}(x^r)$. When $\tau \geq 2$, repeating the local updates for $t$ steps and substituting $\widehat{z}_{i,0}^r = \mathcal{P}_{\mathcal{M}}(x^r)$ and $z_{i,t+1}^r = \mathcal{P}_{\mathcal{M}}(\hat{z}_{i,t+1}^r)$, we have

$$\mathbb{E}\left\|z_{i,t+1}^r - \mathcal{P}_{\mathcal{M}}(x^r)\right\|^2 = \mathbb{E}\left\|\mathcal{P}_{\mathcal{M}}\left(\mathcal{P}_{\mathcal{M}}(x^r) - \eta \sum_{\ell=0}^t \left(\mathrm{grad} f_i(z_{i,\ell}^r; \mathcal{B}_{i,\ell}^r) + c_i^r\right)\right) - \mathcal{P}_{\mathcal{M}}(x^r)\right\|^2. \tag{18}$$

To bound the right-hand side of (18), we compare our algorithm with the exact C-PRGD step given in (7) under the step size $(t+1)\eta$

$$\tilde{x}_{\mathrm{C-PRGD}}^{r+1} := \mathcal{P}_{\mathcal{M}}\left(\mathcal{P}_{\mathcal{M}}(x^r) - (t+1)\eta \mathrm{grad} f(\mathcal{P}_{\mathcal{M}}(x^r))\right).$$

It follows from (15) that

$$\|\tilde{x}_{\mathrm{C-PRGD}}^{r+1} - \mathcal{P}_{\mathcal{M}}(x^r)\| \leq M\tau\eta\|\mathrm{grad} f(\mathcal{P}_{\mathcal{M}}(x^r))\|.$$

Then from (18) we have

$$\mathbb{E}\left\|z_{i,t+1}^r - \mathcal{P}_{\mathcal{M}}(x^r)\right\|^2 \tag{19}$$

$$= \mathbb{E}\left\|\mathcal{P}_{\mathcal{M}}\left(\mathcal{P}_{\mathcal{M}}(x^r) - \eta \sum_{\ell=0}^t \left(\mathrm{grad} f_i(z_{i,\ell}^r; \mathcal{B}_{i,\ell}^r) + c_i^r\right)\right) - \tilde{x}_{\mathrm{C-PRGD}}^{r+1} + \tilde{x}_{\mathrm{C-PRGD}}^{r+1} - \mathcal{P}_{\mathcal{M}}(x^r)\right\|^2$$

$$\leq \underbrace{2\mathbb{E}\left\|\mathcal{P}_{\mathcal{M}}\left(\mathcal{P}_{\mathcal{M}}(x^r) - \eta \sum_{\ell=0}^t (\mathrm{grad} f_i(z_{i,\ell}^r; \mathcal{B}_{i,\ell}^r) + c_i^r)\right) - \tilde{x}_{\mathrm{C-PRGD}}^{r+1}\right\|^2}_{(\mathrm{I})} + 2M^2\tau^2\eta^2\|\mathrm{grad} f(\mathcal{P}_{\mathcal{M}}(x^r))\|^2,$$

where we use $\|a + b\|^2 \leq 2\|a\|^2 + 2\|b\|^2$ in the inequality.

To bound the term (I) on the right hand of (19), from $\tilde{\eta} \leq \frac{\gamma \eta_g}{2D_f}$ and $\max_{i,l,x \in \mathcal{M}} \|\nabla f_{il}(x; \mathcal{D}_{il})\| \leq D_f$, we have

$$\left\|\eta \sum_{\ell=0}^t \left(\mathrm{grad} f_i(z_{i,\ell}^r; \mathcal{B}_{i,\ell}^r) + c_i^r\right)\right\| \leq \gamma.$$

Thus, by substituting definition of $\tilde{x}^{r+1}_{\mathrm{C-PRGD}}$, we can invoke the 2-Lipschitz continuity of $\mathcal{P}_\mathcal{M}$ over $\overline{U}_\mathcal{M}(\gamma)$ given in (3) and get

$$
\begin{aligned}
(\mathrm{I}) &= 2\mathbb{E}\Big\|\mathcal{P}_\mathcal{M}\big(\mathcal{P}_\mathcal{M}(x^r) - \eta\sum_{\ell=0}^{t}\big(\mathrm{grad}f_i(z^r_{i,\ell};\mathcal{B}^r_{i,\ell}) + c^r_i\big)\big) \\
&\quad - \mathcal{P}_\mathcal{M}\big(\mathcal{P}_\mathcal{M}(x^r) - (t+1)\eta\mathrm{grad}f(\mathcal{P}_\mathcal{M}(x^r))\big)\Big\|^2 \\
&\le 4\mathbb{E}\Big\|\eta\sum_{\ell=0}^{t}\big(\mathrm{grad}f_i(z^r_{i,\ell};\mathcal{B}^r_{i,\ell}) + c^r_i - \mathrm{grad}f(\mathcal{P}_\mathcal{M}(x^r))\big)\Big\|^2.
\end{aligned}
\tag{20}
$$

Next, to bound the right-hand side of (20) we rewrite it in terms of $\|\mathbf{\Lambda}^r - \overline{\mathbf{\Lambda}}^r\|^2$ by substituting the definition of the $c^r_i$ given in (5)

$$
(\mathrm{I}) \le 4\mathbb{E}\Big\|\eta\sum_{\ell=0}^{t}\Big(\mathrm{grad}f_i(z^r_{i,\ell};\mathcal{B}^r_{i,\ell}) - \mathrm{grad}f_i(\mathcal{P}_\mathcal{M}(x^r)) \tag{21}
$$

$$
+ \mathrm{grad}f_i(\mathcal{P}_\mathcal{M}(x^r)) + \frac{1}{\tau}\sum_{t=0}^{\tau-1}\frac{1}{n}\sum_{i=1}^{n}\mathrm{grad}f_i(z^{r-1}_{i,t};\mathcal{B}^{r-1}_{i,t})
$$

$$
- \frac{1}{\tau}\sum_{t=0}^{\tau-1}\mathrm{grad}f_i\big(z^{r-1}_{i,t};\mathcal{B}^{r-1}_{i,t}\big) - \mathrm{grad}f(\mathcal{P}_\mathcal{M}(x^r))\Big)\Big\|^2
$$

$$
= 4\mathbb{E}\Big\|\eta\sum_{\ell=0}^{t}\Big(\mathrm{grad}f_i(z^r_{i,\ell};\mathcal{B}^r_{i,\ell}) - \mathrm{grad}f_i(\mathcal{P}_\mathcal{M}(x^r)) + \frac{1}{\eta\tau}\big(\Lambda^r_i - \overline{\Lambda}^r\big)\Big)\Big\|^2
$$

$$
\le \underbrace{8\mathbb{E}\Big\|\eta\sum_{\ell=0}^{t}\big(\mathrm{grad}f_i(z^r_{i,\ell};\mathcal{B}^r_{i,\ell}) - \mathrm{grad}f_i(\mathcal{P}_\mathcal{M}(x^r))\big)\Big\|^2}_{(\mathrm{II})} + 8\mathbb{E}\Big\|\frac{t+1}{\tau}\Lambda^r_i - \frac{t+1}{\tau}\overline{\Lambda}^r\Big\|^2.
$$

Next, for the term (II), substituting Assumption 4.2 yields

$$
(\mathrm{II}) = 8\mathbb{E}\Big\|\eta\sum_{\ell=0}^{t}\big(\mathrm{grad}f_i(z^r_{i,\ell};\mathcal{B}^r_{i,\ell}) - \mathrm{grad}f_i(z^r_{i,\ell}) + \mathrm{grad}f_i(z^r_{i,\ell}) - \mathrm{grad}f_i(\mathcal{P}_\mathcal{M}(x^r))\big)\Big\|^2 \tag{22}
$$

$$
\le 16(t+1)^2\mathbb{E}\Big\|\frac{\eta}{t+1}\sum_{\ell=0}^{t}\big(\mathrm{grad}f_i(z^r_{i,\ell};\mathcal{B}^r_{i,\ell}) - \mathrm{grad}f_i(z^r_{i,\ell})\big)\Big\|^2
$$

$$
+ 16\mathbb{E}\Big\|\eta\sum_{\ell=0}^{t}\big(\mathrm{grad}f_i(z^r_{i,\ell}) - \mathrm{grad}f_i(\mathcal{P}_\mathcal{M}(x^r))\big)\Big\|^2.
$$

For the first term can be handled by the fact [48, Corollary C.1] that

$$
16(t+1)^2\eta^2\mathbb{E}\Big\|\frac{1}{t+1}\sum_{\ell=0}^{t}\big(\mathrm{grad}f_i(z^r_{i,\ell};\mathcal{B}^r_{i,\ell}) - \mathrm{grad}f_i(z^r_{i,\ell})\big)\Big\|^2 \tag{23}
$$

$$
= \frac{16(t+1)^2\eta^2}{(t+1)^2}\sum_{\ell=0}^{t}\mathbb{E}\Big[\mathbb{E}\big[\|\big(\mathrm{grad}f_i\big(z^r_{i,\ell};\mathcal{B}^r_{i,\ell}\big) - \mathrm{grad}f_i\big(z^r_{i,\ell}\big)\|^2|\mathcal{F}^r_t\big]\Big] \le \frac{1}{t+1}\frac{\sigma^2}{b}.
$$

Combining (22), (23), and (21), we have

$$
(\mathrm{I}) \le 16(t+1)\eta^2 L^2\sum_{\ell=0}^{t}\mathbb{E}\|z^r_{i,\ell} - \mathcal{P}_\mathcal{M}(x^r)\|^2 + 8\Big(\frac{t+1}{\tau}\Big)^2\mathbb{E}\|\Lambda^r_i - \overline{\Lambda}^r\|^2 + 16(t+1)\eta^2\frac{\sigma^2}{b}, \tag{24}
$$

where we use Assumption 4.1. Next we substitute (24) into (19) to get

$$
\mathbb{E}[\|z^r_{i,t+1} - \mathcal{P}_\mathcal{M}(x^r)\|^2] \tag{25}
$$

$$\leq 16(t+1)\eta^2 L^2 \sum_{\ell=0}^{t} \mathbb{E}\|z_{i,\ell}^r - \mathcal{P}_{\mathcal{M}}(x^r)\|^2$$

$$+ \underbrace{2M^2\tau^2\eta^2\|\mathrm{grad}f(\mathcal{P}_{\mathcal{M}}(x^r))\|^2 + 8\mathbb{E}\|\Lambda_i^r - \overline{\Lambda}^r\|^2 + 16\tau\eta^2\frac{\sigma^2}{b}}_{:=A^r}.$$

The following proof is similar to that in [24]. We define $A^r$ as the sum of the last three terms on the right hand of (25) and $S_{i,t}^r := \sum_{\ell=0}^{t} \mathbb{E}\|z_{i,\ell}^r - \mathcal{P}_{\mathcal{M}}(x^r)\|^2$. By $\mathbb{E}[\|z_{i,t+1}^r - \mathcal{P}_{\mathcal{M}}(x^r)\|^2] = S_{i,t+1}^r - S_{i,t}^r$ and (25), we have

$$S_{i,t+1}^r \leq (1 + 1/(16\tau)) S_{i,t}^r + A^r, \tag{26}$$

where the inequality is from $\tilde{\eta} \leq \eta_g/(16L)$ and thus $16(t+1)\eta^2 L^2 \leq 1/(16\tau)$. With (26), we get

$$S_{i,\tau-1}^r \leq A^r \sum_{\ell=0}^{\tau-2} (1 + 1/(16\tau))^\ell \leq 1.1\tau A^r, \tag{27}$$

where we use $\sum_{\ell=0}^{\tau-2} (1 + 1/(16\tau))^\ell \leq \sum_{\ell=0}^{\tau-2} \exp(\ell/(16\tau)) \leq \sum_{\ell=0}^{\tau-2} \exp(1/16) \leq 1.1\tau$. Summing (27) over all the clients $i$, we get

$$\mathbb{E}\Big[\sum_{i=1}^{n}\sum_{t=0}^{\tau-1} \|z_{i,t}^r - \mathcal{P}_{\mathcal{M}}(x^r)\|^2\Big] \tag{28}$$

$$\leq 3nM^2\tau^3\eta^2\|\mathrm{grad}f(\mathcal{P}_{\mathcal{M}}(x^r))\|^2 + 9\tau\mathbb{E}\|\Lambda^r - \overline{\Lambda}^r\|^2 + 18n\tau^2\eta^2\frac{\sigma^2}{b}.$$

Now we are ready to bound $\frac{1}{n}\mathbb{E}\|\Lambda^{r+1} - \overline{\Lambda}^{r+1}\|^2$. By the definition of $\Lambda^{r+1}$ and $\overline{\Lambda}^{r+1}$ we have

$$\mathbb{E}\|\Lambda^{r+1} - \overline{\Lambda}^{r+1}\|^2 \tag{29}$$

$$= \eta^2\mathbb{E}\Big\|\tau\mathrm{gradf}\left(\mathcal{P}_{\mathcal{M}}(\mathbf{x}^{r+1})\right) - \sum_{t=0}^{\tau-1} \mathrm{gradf}\left(\mathbf{z}_t^r; \mathcal{B}_t^r\right) - \tau\overline{\mathrm{gradf}}(\mathcal{P}_{\mathcal{M}}(\mathbf{x}^{r+1})) + \sum_{t=0}^{\tau-1} \overline{\mathrm{gradf}}\left(\mathbf{z}_t^r; \mathcal{B}_t^r\right)\Big\|^2$$

$$\leq \eta^2\mathbb{E}\Big\|\tau\mathrm{gradf}\left(\mathcal{P}_{\mathcal{M}}(\mathbf{x}^{r+1})\right) - \sum_{t=0}^{\tau-1} \mathrm{gradf}\left(\mathbf{z}_t^r; \mathcal{B}_t^r\right)\Big\|^2$$

$$= \eta^2\mathbb{E}\Big\|\tau\mathrm{gradf}\left(\mathcal{P}_{\mathcal{M}}(\mathbf{x}^{r+1})\right) - \tau\mathrm{gradf}\left(\mathcal{P}_{\mathcal{M}}(\mathbf{x}^r)\right) + \tau\mathrm{gradf}\left(\mathcal{P}_{\mathcal{M}}(\mathbf{x}^r)\right)$$

$$- \sum_{t=0}^{\tau-1} \mathrm{gradf}\left(\mathbf{z}_t^r\right) + \sum_{t=0}^{\tau-1} \mathrm{gradf}\left(\mathbf{z}_t^r\right) - \sum_{t=0}^{\tau-1} \mathrm{gradf}\left(\mathbf{z}_t^r; \mathcal{B}_t^r\right)\Big\|^2$$

$$\leq 2\eta^2\tau^2 L^2 n\mathbb{E}\left\|\mathcal{P}_{\mathcal{M}}(\mathbf{x}^{r+1}) - \mathcal{P}_{\mathcal{M}}(\mathbf{x}^r)\right\|^2 + 4\eta^2\tau L^2\sum_{i=1}^{n}\sum_{t=0}^{\tau-1} \mathbb{E}\left\|z_{i,t}^r - \mathcal{P}_{\mathcal{M}}(x^r)\right\|^2 + 4\eta^2\tau n\frac{\sigma^2}{b}.$$

Here, the first inequality is due to $\|\Lambda^{r+1} - \overline{\Lambda}^{r+1}\|^2 \leq \|\Lambda^{r+1}\|^2$, the last inequality is due to $\|a+b\|^2 \leq 2\|a\|^2 + 2\|b\|^2$, Assumption 4.1, and following similar derivations as in (23).

By substituting (28) into (29) and reorganizing the results, we complete the proof of Lemma A.3. $\square$

## A.3 Proof of Theorem 4.3

To bound the first term in the Lyapunov function, we focus on the server-side update. We begin with the following lemma over the manifolds.

**Lemma A.4.** *Given $x \in \mathcal{M}$, $v \in T_x\mathcal{M}$, $\eta > 0$, $x - \eta v \in \overline{U}_{\mathcal{M}}(\gamma)$, and $x^+ = \mathcal{P}_{\mathcal{M}}(x - \eta v)$, it holds that*

$$f(x^+) \leq f(z) + \langle \mathrm{grad}f(x) - v, x^+ - z \rangle$$

$$- \frac{1}{2\eta}(\|x^+ - x\|^2 - \|z - x\|^2) - \left(\frac{1}{2\eta} - \frac{3\|v\|}{4\gamma}\right)\|z - x^+\|^2 \tag{30}$$

$$+ \frac{L}{2}\|x^+ - x\|^2 + \frac{L}{2}\|z - x\|^2, \forall z \in \mathcal{M}.$$

*Proof.* For any $\mu$-strongly convex function $h$, we have for any $y, z \in \mathcal{M}$

$$h(z) \geq h(y) + \langle \nabla h(y), z - y \rangle + \frac{\mu}{2} \|z - y\|^2$$

$$= h(y) + \langle \mathrm{grad}h(y) + \nabla h(y) - \mathrm{grad}h(y), z - y \rangle + \frac{\mu}{2} \|z - y\|^2 \tag{31}$$

$$\geq h(y) + \langle \mathrm{grad}h(y), z - y \rangle + \left( \frac{\mu}{2} - \frac{\|\nabla h(y)\|}{4\gamma} \right) \|z - y\|^2,$$

where the second inequality is from the normal inequality (4) and $\|\nabla h(y) - \mathrm{grad}h(y)\| \leq \|\nabla h(y)\|$. Setting $h(y) = \frac{1}{2\eta} \|y - (x - \eta v)\|^2$ in (31) with $\mu = 1/\eta$, $y = x^+$, and noting the optimality of $x^+$ (i.e., $\mathrm{grad}h(x^+) = 0$), we have

$$\frac{1}{2\eta} \|z - (x - \eta v)\|^2 \geq \frac{1}{2\eta} \|x^+ - (x - \eta v)\|^2 + \left( \frac{1}{2\eta} - \frac{\|y - (x - \eta v)\|}{4\eta\gamma} \right) \|z - x^+\|^2$$

$$\geq \frac{1}{2\eta} \|x^+ - (x - \eta v)\|^2 + \left( \frac{1}{2\eta} - \frac{3\|v\|}{4\gamma} \right) \|z - x^+\|^2,$$

where the second inequality is from $x - \eta v \in \overline{U}_{\mathcal{M}}(\gamma)$ and $\|\mathcal{P}_{\mathcal{M}}(x - \eta v) - (x - \eta v)\| \leq \|\mathcal{P}_{\mathcal{M}}(x - \eta v) - x\| + \eta\|v\| \leq 3\eta\|v\|$. Rearranging the above inequality leads to

$$\langle v, z - x^+ \rangle \geq \frac{1}{2\eta} (\|x^+ - x\|^2 - \|z - x\|^2) + \left( \frac{1}{2\eta} - \frac{3\|v\|}{4\gamma} \right) \|z - x^+\|^2. \tag{32}$$

It follows from the $L$-smoothness of $f$ that

$$f(x^+) \leq f(x) + \langle \mathrm{grad}f(x), x^+ - x \rangle + \frac{L}{2} \|x^+ - x\|^2$$

$$\leq f(z) + \langle \mathrm{grad}f(x), x^+ - z \rangle + \frac{L}{2} \|z - x\|^2 + \frac{L}{2} \|x^+ - x\|^2,$$

where we use $f(x) + \langle \mathrm{grad}f(x), z - x \rangle - \frac{L}{2} \|z - x\|^2 \leq f(z)$ in the last inequality. Combining the above inequality and (32) gives (30). $\qquad \square$

In the following, we use Lemma A.4 to (7) and (6), respectively. First, to apply Lemma A.4 to (7), we substitute $x^+ = \tilde{x}^{r+1}$, $z = \mathcal{P}_{\mathcal{M}}(x^r)$, $x = \mathcal{P}_{\mathcal{M}}(x^r)$, and $v = \mathrm{grad}f(\mathcal{P}_{\mathcal{M}}(x^r))$ and get

$$\mathbb{E}\left[ f\left( \tilde{x}^{r+1} \right) \right] \leq \mathbb{E}\left[ f\left( \mathcal{P}_{\mathcal{M}}(x^r) \right) + \left( \frac{L}{2} - \frac{1}{2\tilde{\eta}} \right) \left\| \tilde{x}^{r+1} - \mathcal{P}_{\mathcal{M}}(x^r) \right\|^2 - \frac{1 - \tilde{\eta}\rho}{2\tilde{\eta}} \left\| \tilde{x}^{r+1} - \mathcal{P}_{\mathcal{M}}(x^r) \right\|^2 \right],$$

$$\tag{33}$$

where we use $\tilde{\eta} \leq \frac{\gamma}{D_f}$ to guarantee $\tilde{x}^{r+1} \in \overline{U}_{\mathcal{M}}(\gamma)$ and $\rho := \frac{3D_f}{2\gamma}$.

Next, to use Lemma A.4 to (6), we set $x^+ = \mathcal{P}_{\mathcal{M}}(x^{r+1})$, $x = \mathcal{P}_{\mathcal{M}}(x^r)$, $z = \tilde{x}^{r+1}$, and $v = v^r$ and get

$$\mathbb{E}\left[ f(\mathcal{P}_{\mathcal{M}}(x^{r+1})) \right] \tag{34}$$

$$\leq \mathbb{E}\Big[ f(\tilde{x}^{r+1}) + \langle \mathrm{grad}f(\mathcal{P}_{\mathcal{M}}(x^r)) - v^r, \mathcal{P}_{\mathcal{M}}(x^{r+1}) - \tilde{x}^{r+1} \rangle$$

$$- \frac{1}{2\tilde{\eta}} \left( \|\mathcal{P}_{\mathcal{M}}(x^{r+1}) - \mathcal{P}_{\mathcal{M}}(x^r)\|^2 - \|\tilde{x}^{r+1} - \mathcal{P}_{\mathcal{M}}(x^r)\|^2 \right) - \frac{1 - \tilde{\eta}\rho}{2\tilde{\eta}} \|\tilde{x}^{r+1} - \mathcal{P}_{\mathcal{M}}(x^{r+1})\|^2$$

$$+ \frac{L}{2} \left\| \mathcal{P}_{\mathcal{M}}(x^{r+1}) - \mathcal{P}_{\mathcal{M}}(x^r) \right\|^2 + \frac{L}{2} \|\tilde{x}^{r+1} - \mathcal{P}_{\mathcal{M}}(x^r)\|^2 \Big]$$

$$= \mathbb{E}\Big[ f\left( \tilde{x}^{r+1} \right) + \langle \mathrm{grad}f(\mathcal{P}_{\mathcal{M}}(x^r)) - v^r, \mathcal{P}_{\mathcal{M}}(x^{r+1}) - \tilde{x}^{r+1} \rangle + \left( \frac{L}{2} - \frac{1}{2\tilde{\eta}} \right) \left\| \mathcal{P}_{\mathcal{M}}(x^{r+1}) - \mathcal{P}_{\mathcal{M}}(x^r) \right\|^2$$

$$+ \left( \frac{L}{2} + \frac{1}{2\tilde{\eta}} \right) \left\| \tilde{x}^{r+1} - \mathcal{P}_{\mathcal{M}}(x^r) \right\|^2 - \frac{1 - \tilde{\eta}\rho}{2\tilde{\eta}} \left\| \mathcal{P}_{\mathcal{M}}(x^{r+1}) - \tilde{x}^{r+1} \right\|^2 \Big],$$

where we use $\tilde{\eta} \leq \frac{\gamma\eta_g}{2D_f}$ to guarantee $\mathcal{P}_{\mathcal{M}}(x^{r+1}) \in \overline{U}_{\mathcal{M}}(\gamma)$.

Recall that

$$\mathcal{P}_\mathcal{M}(x^{r+1}) = \mathcal{P}_\mathcal{M}\Big(\mathcal{P}_\mathcal{M}(x^r) - \tilde\eta \underbrace{\frac{1}{n\tau}\sum_{i=1}^n \sum_{t=0}^{\tau-1}\big(\mathrm{grad}f_i\left(z_{i,t}^r;\mathcal{B}_{i,t}^r\right)\big)}_{:=v^r}\Big).$$

Combining the above inequality, (33), and (34) yields

$$\mathbb{E}\left[f(\mathcal{P}_\mathcal{M}(x^{r+1}))\right] \tag{35}$$
$$\leq \mathbb{E}\big[f\left(\mathcal{P}_\mathcal{M}(x^r)\right) + \left(L - \frac{1}{2\tilde\eta} + \frac{\rho}{2}\right)\|\tilde x^{r+1} - \mathcal{P}_\mathcal{M}(x^r)\|^2 + \left(\frac{L}{2} - \frac{1}{2\tilde\eta}\right)\|\mathcal{P}_\mathcal{M}(x^{r+1}) - \mathcal{P}_\mathcal{M}(x^r)\|^2$$
$$\underbrace{-\frac{1-\tilde\eta\rho}{2\tilde\eta}\|\mathcal{P}_\mathcal{M}(x^{r+1}) - \tilde x^{r+1}\|^2}_{(\mathrm{IV})} + \underbrace{\left\langle \mathcal{P}_\mathcal{M}(x^{r+1}) - \tilde x^{r+1}, \mathrm{grad}f\left(\mathcal{P}_\mathcal{M}(x^r)\right) - v^r\right\rangle}_{(\mathrm{V})}\big].$$

According to $\|a+b\|^2 \leq \frac{1}{2\tilde\eta}\|a\|^2 + \frac{\tilde\eta}{2}\|b\|^2$, we have

$$(\mathrm{IV}) + (\mathrm{V}) \tag{36}$$
$$\leq (\mathrm{IV}) + \frac{1-\tilde\eta\rho}{2\tilde\eta}\|\mathcal{P}_\mathcal{M}(x^{r+1}) - \tilde x^{r+1}\|^2 + \frac{\tilde\eta}{2(1-\tilde\eta\rho)}\|\mathrm{grad}f(\mathcal{P}_\mathcal{M}(x^r)) - v^r\|^2$$
$$= \frac{\tilde\eta}{2(1-\tilde\eta\rho)}\|\mathrm{grad}f(\mathcal{P}_\mathcal{M}(x^r)) - v^r\|^2.$$

To bound the above inequality, following similar derivations as in (23) we obtain

$$\mathbb{E}\left\|v^r - \mathrm{grad}f(\mathcal{P}_\mathcal{M}(x^r))\right\|^2$$
$$= \mathbb{E}\Big\|\frac{1}{n\tau}\sum_{i=1}^n\sum_{t=0}^{\tau-1}\Big(\mathrm{grad}f_i\left(z_{i,t}^r;\mathcal{B}_{i,t}^r\right) - \mathrm{grad}f_i(z_{i,t}^r) + \mathrm{grad}f_i(z_{i,t}^r) - \mathrm{grad}f_i(\mathcal{P}_\mathcal{M}(x^r))\Big)\Big\|^2 \tag{37}$$
$$\leq 2L^2\frac{1}{n\tau}\sum_{i=1}^n\sum_{t=0}^{\tau-1}\mathbb{E}\|z_{i,t}^r - \mathcal{P}_\mathcal{M}(x^r)\|^2 + \frac{2}{\tau n}\frac{\sigma^2}{b}.$$

Substituting (36) and (37) into (35), we have

$$\mathbb{E}[f(\mathcal{P}_\mathcal{M}(x^{r+1}))] \tag{38}$$
$$\leq \mathbb{E}\Big[f\left(\mathcal{P}_\mathcal{M}(x^r)\right) + \left(L - \frac{1}{2\tilde\eta} + \frac{\rho}{2}\right)\|\tilde x^{r+1} - \mathcal{P}_\mathcal{M}(x^r)\|^2 + \left(\frac{L}{2} - \frac{1}{2\tilde\eta}\right)\|\mathcal{P}_\mathcal{M}(x^{r+1}) - \mathcal{P}_\mathcal{M}(x^r)\|^2$$
$$+ \frac{\tilde\eta}{2(1-\tilde\eta\rho)}\left(\frac{2L^2}{n\tau}\sum_{i=1}^n\sum_{t=0}^{\tau-1}\|z_{i,t}^r - \mathcal{P}_\mathcal{M}(x^r)\|^2 + \frac{2}{\tau n}\frac{\sigma^2}{b}\right)\Big].$$

The final term is the drift-error that can be bounded in (28). Thus, (38) becomes

$$\mathbb{E}[f(\mathcal{P}_\mathcal{M}(x^{r+1}))] \tag{39}$$
$$\leq \mathbb{E}\Big[f\left(\mathcal{P}_\mathcal{M}(x^r)\right) + \left(L - \frac{1}{2\tilde\eta} + \frac{\rho}{2}\right)\|\tilde x^{r+1} - \mathcal{P}_\mathcal{M}(x^r)\|^2 + \left(\frac{L}{2} - \frac{1}{2\tilde\eta}\right)\|\mathcal{P}_\mathcal{M}(x^{r+1}) - \mathcal{P}_\mathcal{M}(x^r)\|^2$$
$$+ \frac{\tilde\eta}{2(1-\tilde\eta\rho)}\frac{2L^2}{n\tau}\left(12nM^2\tau^3\eta^2\|\mathcal{G}_{\tilde\eta g}(\mathcal{P}_\mathcal{M}(x^r))\|^2 + 9\tau\mathbb{E}\|\boldsymbol{\Lambda}^r - \overline{\boldsymbol{\Lambda}}^r\|^2 + 18n\tau^2\eta^2\frac{\sigma^2}{b}\right) + \frac{\tilde\eta}{2(1-\tilde\eta\rho)}\frac{2}{\tau n}\frac{\sigma^2}{b}\Big],$$

where we use $\|\mathrm{grad}f(\mathcal{P}_{\mathcal{M}(x^r)})\| \leq 2\|\mathcal{G}_{\tilde\eta}(\mathcal{P}_{\mathcal{M}(x^r)})\|$ from Lemma (A.2). By substituting $\tilde\eta \leq \frac{1}{4\rho}$ as $\tilde\eta \leq \frac{\gamma}{6D_f}$, we have $\frac{\tilde\eta}{2(1-\tilde\eta\rho)} \leq \tilde\eta$. Combining the recursions given by Lemma A.3 and (39), we have for the Lyapunov function that

$$\mathbb{E}\Big[\left(f(\mathcal{P}_\mathcal{M}(x^{r+1})) - f^\star\right) + \frac{1}{\tilde\eta n}\|\boldsymbol{\Lambda}^{r+1} - \overline{\boldsymbol{\Lambda}}^{r+1}\|^2\Big] \tag{40}$$

$$\leq \mathbb{E}\big[\, (f(\mathcal{P}_{\mathcal{M}}(x^r)) - f^\star) + \frac{1}{\tilde{\eta} n}\|\mathbf{\Lambda}^r - \overline{\mathbf{\Lambda}}^r\|^2 - \frac{\tilde{\eta}}{8}\,\|\mathcal{G}(\mathcal{P}_{\mathcal{M}}(x^r))\|^2 \big] + \frac{8\tilde{\eta}}{n\tau}\frac{\sigma^2}{b},$$

where we substitute conditions (11) on the step sizes and omit straightforward algebraic calculations. Substituting the definition of the Lyapunov function $\Omega^r$ and repeating the above inequality, we complete the proof of Theorem 4.3.

**Discussion on step sizes.** In our analysis, the condition $\eta_g = \sqrt{n}$ is applied only in the final step, when combining Lemma A.3 and (39) to derive the recursion on the Lyapunov function in (40). Up to this point, we retain $\eta_g$ as a variable without assigning a specific value, ensuring that all preceding results remain valid for $\eta_g = 1$ as well. We choose $\eta_g = \sqrt{n}$ in (40) to conveniently cancel out the number of clients $n$, yielding a more concise expression in the final result.

In (11), we require that $\tilde{\eta} := \eta_g \eta \tau \leq \min\left\{\frac{1}{24ML}, \frac{\gamma}{6D_f}, \frac{1}{D_f L_{\mathcal{P}}}\right\}$. The term $\frac{\gamma}{6D_f}$ controls the client drift, see (39); the term $\frac{1}{D_f L_{\mathcal{P}}}$ is used to derive (16), which ensures that when the metric $\|\mathcal{G}_{\tilde{\eta}}(\mathcal{P}_{\mathcal{M}}(x^r))\|$ approaches zero, the first-order optimality condition is met. The term $\frac{1}{24ML}$ is used to establish the Lyapunov function recursion (40) by combining the recursions in (17) and (39). Additional conditions on the step sizes ensure that the iterates remain within the $2\gamma$-tube during our analysis, such as in (17), (33), and (34), but they are implied by the three terms in (11).

### A.4 Additional results for numerical experiments

#### A.4.1 kPCA

**The settings for kPCA problem with Mnist Dataset.** The Mnist dataset consists of 60,000 handwritten digit images ranging from 0 to 9, each with dimensions of $28 \times 28$. We reshape these images into a data matrix $A \in \mathbb{R}^{60000 \times 784}$. To construct the heterogeneous $A_i$, we sort the rows in increasing order of their associated digits and then split every $60000/n$ rows, with $n = 10$ as the number of clients, among each client. This data partition introduces significant data heterogeneity. In our setup, $d = 784$, $p = 6000$, and $k = 2$.

For kPCA problem with Mnist dataset, when $\tau = 10$, the comparison on $f(x^r) - f^\star$ is shown in Figs. 5. Our algorithm performs much better than alternative methods in terms of communication quantity and run time.

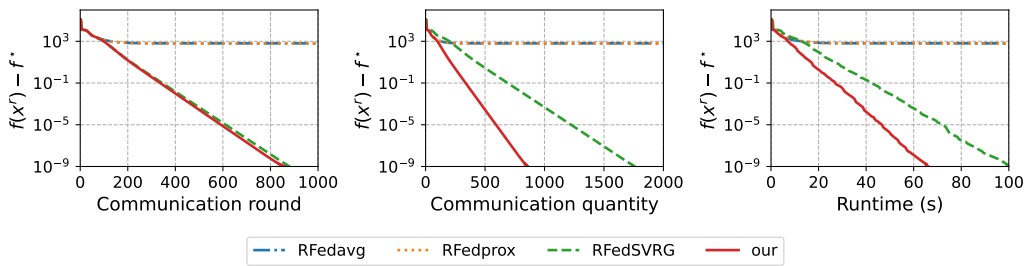

Figure 5: kPCA problem with Mnist dataset: Comparison on $f(x^r) - f^\star$.

**Synthetic Dataset.** We also solve kPCA with synthetic datasets on larger networks with $n = 30$. We generate each entry of $A_i$ from Gaussian distribution $\mathcal{N}\left(0, \frac{2i}{n}\right)$ such that $A_i$ are heterogeneous among clients. We set $(d, k) = (20, 5)$ and $p = 15$. We use the local full gradient $\nabla f_i$ to remove the influence of stochastic Riemannian gradient noise.

In the first set of experiments, we compare with existing algorithms, including RFedavg, RFedprox, and RFedSVRG. For all algorithms, we set the number of local steps as $\tau = 5$ and the step size as $\eta = 4e{-}3$. For our algorithm, we set $\eta_g = 1$. The experimental results are shown in Figs. 6. The $y$-axis represents $\|\text{grad} f(x^r)\|$ and $(f(x^r) - f^\star)$ respectively, while the $x$-axis represents the number of communication rounds, communication quantity, and run time, respectively. It can be observed that RFedavg and RFedprox face the issue of client drift, hence they do not converge accurately. Both FedSVRG and our algorithm can overcome the client drift issue, but our algorithm is slightly faster

in terms of communication rounds and is much faster in terms of both communication quantity and run time.

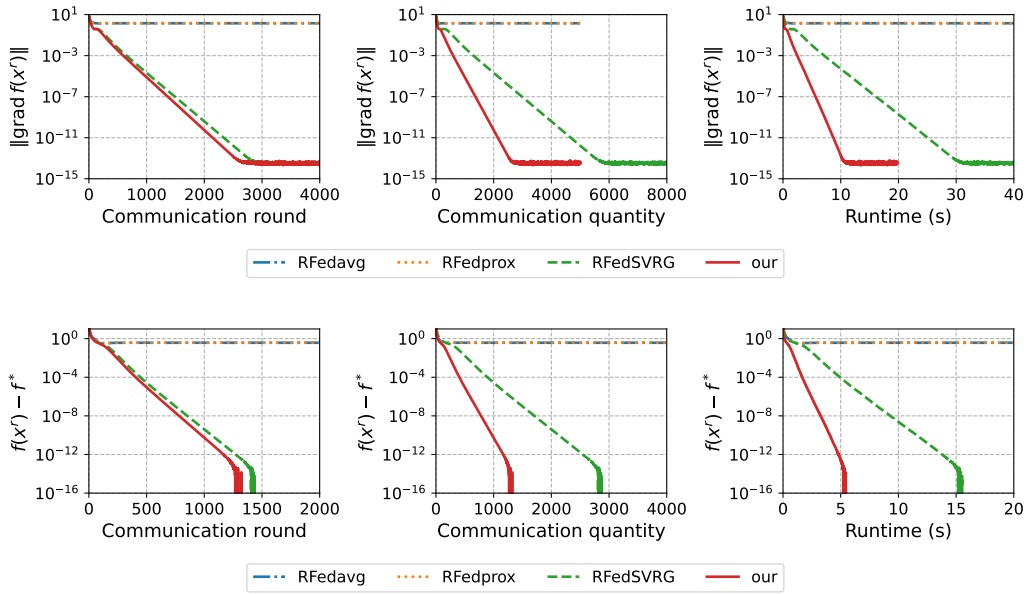

Figure 6: kPCA with synthetic dataset: Comparison on $\|\mathrm{grad} f(x^r)\|$ and $f(x^r) - f^\star$.

In the second set of experiments, we test the impact of the number of local updates $\tau$. For all the algorithms, we set $R = 4000$, the step size $\eta = 0.7e-3$, and $\tau \in \{10, 15, 20\}$. For our algorithm, we set $\eta_g = 1$. The results are shown in Figs. 7, with the $y$-axis representing $\|\mathrm{grad} f(x^r)\|$ and $x$-axis representing the communication quantity. When $\tau$ increases, the convergence becomes faster. For all values of $\tau$, our algorithm achieves high accuracy and requires less time.

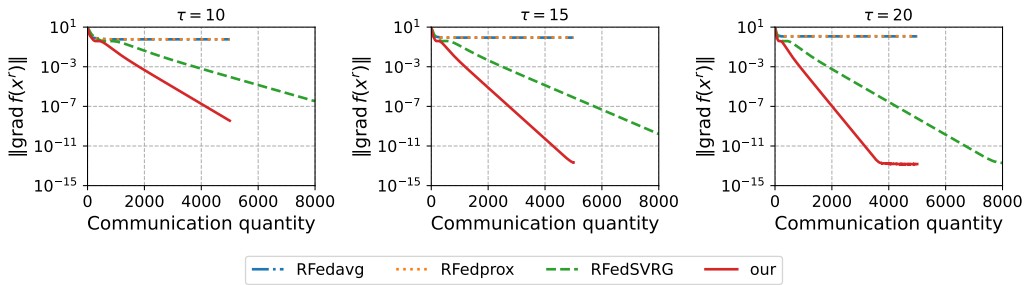

Figure 7: kPCA with synthetic dataset: The impacts of $\tau$.

### A.4.2 Low-rank matrix completion

For numerical tests, we consider random generated $A$. To be specific, we first generate two random matrices $\hat{L} \in \mathbb{R}^{d \times k}$ and $\hat{R} \in \mathbb{R}^{k \times T}$, where each entry obeys the standard Gaussian distribution. For the indices set $\Omega$, we generate a random matrix $B$ with each entry following from the uniform distribution, then set $\Omega_{ij} = 1$ if $B_{ij} \le \nu$ and 0 otherwise. The parameter $\nu$ is set to $10k(d + T - k)/(dT)$.

As shown in Figs. 8, our algorithm is faster than existing algorithms in terms of communication quantity and run time.

We also show the impact of $\tau$ and the impact of $n$. As shown in Figs. 9, larger $\tau$ yields less communication quantity to achieve the same accuracy. As shown in Figs. 10, for $n = 10$, $n = 20$, and $n = 30$, our algorithm consistently outperforms the compared algorithms.

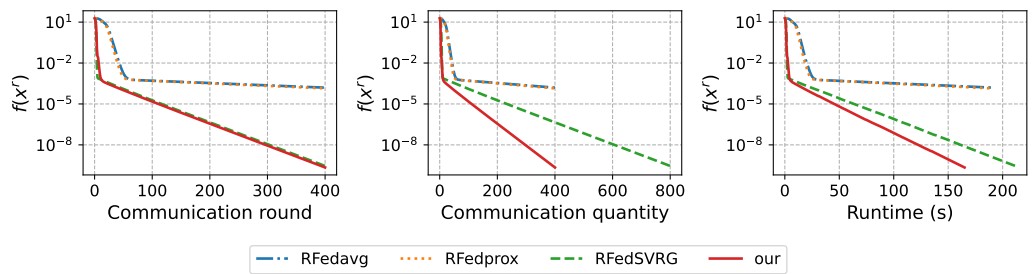

Figure 8: LRMC: Comparison on $f(x^r)$.

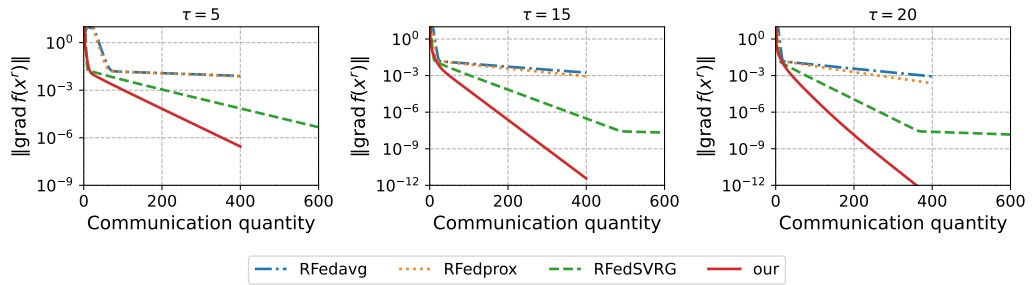

Figure 9: LRMC: The impacts of $\tau$.

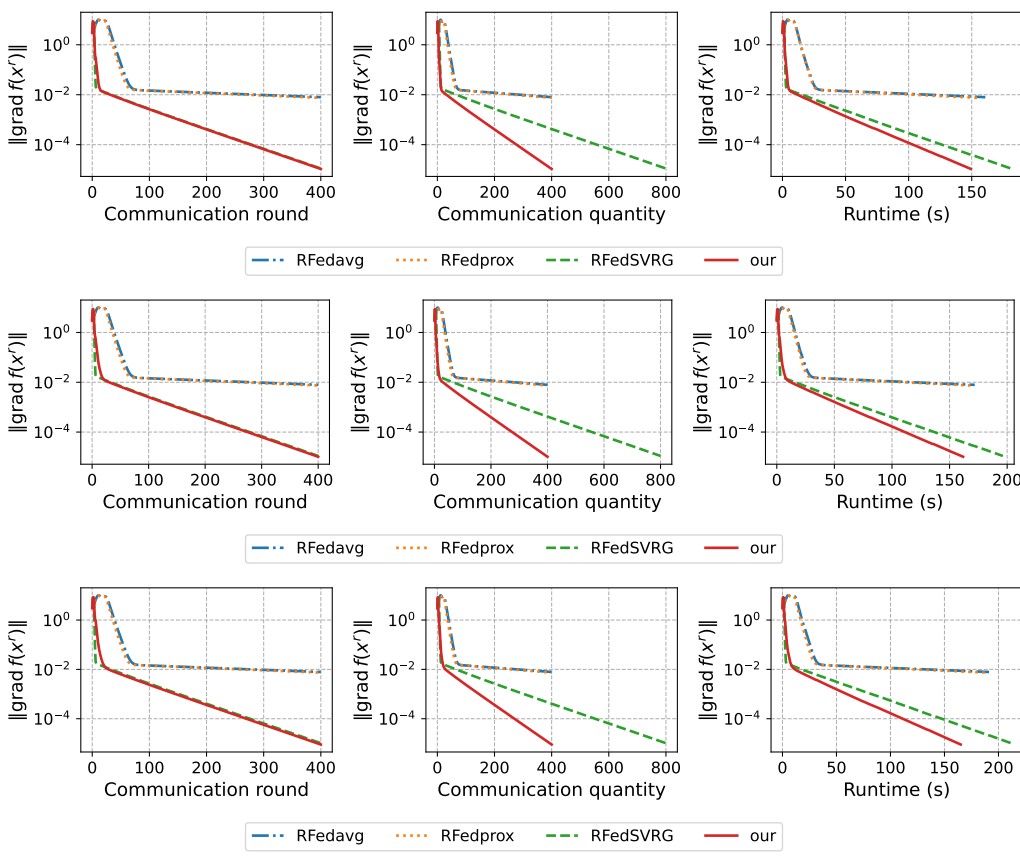

Figure 10: LRMC: The impacts of $n$. (First row: $n = 10$, second row: $n = 20$, and third row: $n = 30$. We set $\tau = 5$. )

