# OpenReview forum: "Nonconvex Federated Learning on Compact Smooth Submanifolds With Heterogeneous Data"
_NeurIPS.cc/2024/Conference — NeurIPS 2024 poster_

### Official Review · Reviewer_3dV9 · 2024-07-12

**Soundness:** 2
**Presentation:** 2
**Contribution:** 2
**Rating:** 4
**Confidence:** 5

**Summary:**

This paper studies federated learning on a compact smooth submanifold using distributed Riemannian optimization methods.

The proposed methods were examined on PCA and low-rank matrix completion tasks.

**Strengths:**

The proposed proximal smoothness of a manifold is intuitive and interesting. The notation can be improved but the theoretical analyses of the algorithm are clear in general.

**Weaknesses:**

In the theoretical analyses, a comparison with the other related Euclidean and Riemannian methods can be given.

Experimental analyses are limited and should be improved as well.

**Questions:**

There are several other works on distributed Riemannian optimization methods for similar tasks, such as
Distributed Principal Component Analysis with Limited Communication, Neurips 2021.
Communication-Efficient Distributed PCA by Riemannian Optimization, ICML 2020.

Could you please compare your proposed methods with these related work theoretically and/or experimentally?

The number of clients considered in the experimental analyses is pretty low. Could you please examine how the results change with different larger number of clients?

How did you measure CPU time?

Could you please provide comparative analyses with Euclidean baselines, such as FedAvg and FedProx?

In FL, a major issue is privacy, and therefore, several mechanisms, such as differential privacy, are utilized. Could you please provide analyses with different privacy mechanisms?

One of the major claims of the paper is improvement of the results compared to the baseline for optimization with heterogeneous data. However, there are not detailed analyses justifying this claim. Could you please provide additional analyses exploring the methods for different data heterogeneity?

**Limitations:**

Limitations were partially addressed.

---

> ### Author Rebuttal · Authors · 2024-08-06
>
> Thank you very much for your careful review and valuable comments.
>
> Weaknesses
>
> We have addressed all the comments accordingly. Please see the response to Questions for details.
>
> Questions
>
> 1) Thank you very much for your comments. We have provided a comparison with [ICML 2020] and [Neurips 2021] in the general response.
>
> 2) Thank you very much for your suggestion.  In our kPCA experiments, we presented results for the MNIST dataset with $n=10$ clients in the main text and for synthetic datasets with $n=30$ clients in the appendix. The results for kPCA on MNIST with $n=30$ clients, reported in Figs.6-7 in the Appendix, are qualitatively similar to those observed for $n=10$, demonstrating that our algorithm effectively reduces communication quantity and runtime. In the MNIST experiment, when we set $n=10$, data from each of the 10 classes was distributed across 10 clients, ensuring that no two clients shared data with the same label. This setup introduced significant data heterogeneity, allowing us to better demonstrate our algorithm's ability to overcome client drift issues caused by such a high level of heterogeneity.
>
> Based on your feedback, we have conducted additional experiments on low-rank matrix completion for $n=10, 20, 30$. In all cases, our algorithm consistently outperforms the compared algorithms. The results are in the uploaded PDF in the general response.
>
> 3) Thank you for raising this issue. We should indeed use "runtime" instead of  "CPU time." Our code was implemented in Python, and to measure runtime we recorded timestamps at the start and end of each communication round using time.time( ). The difference between these two timestamps represents the runtime for one communication round.
>
> 4) Thank you very much for your suggestion. We have provided the comparison with FedAvg and FedProx.  Please see the general response.
>
> 5) This is an excellent question. Our algorithm can indeed be extended to achieve differential privacy. Below, we outline how this can be accomplished.
>
> Algorithmically, each client can add Gaussian noise to the local gradient information to ensure differential privacy protection. Let $\xi \in \mathbb{R}^{d\times k}$ be Gaussian noise, where each entry is independently and identically distributed, following $\mathcal{N}(0, \sigma^2)$. We modify Algorithm 1 at Line 8 and Line 17 as follows:
> $$
> \begin{aligned}
> \widehat{z}_{i,{t+1}}^r&=\widehat{z}\_{i,t}^r   -\eta (\widetilde{\operatorname{grad}}f_i (z\_{i,t}^r;\mathcal{B}\_{i,t}^r)+c_i^r )\\\\
> c_i^{r+1}&=\frac{1}{\eta_g \eta\tau} (\mathcal{P}\_\mathcal{M} (x^r)-x^{r+1})-\frac{1}{\tau} \sum\_{t=0}^{\tau-1} \widetilde{\operatorname{grad}} f_i(z\_{i,t}^r;\mathcal{B}\_{i,t}^r)
> \end{aligned}
> $$
> where $\widetilde{\operatorname{grad}} f_i(z\_{i,t}^r;\mathcal{B}\_{i,t}^r):=\operatorname{grad} f_i(z\_{i,t}^r;\mathcal{B}\_{i,t}^r) +\xi $.
>
> For the analysis, we can establish a quantitative relationship between privacy loss and the variance $ \sigma^2$ as follows:
>
> i) First, we determine the privacy loss for a single local update. According to the post-processing theorem [Lemma 1.8, TCC 2016] and the definition of $\rho$-zCDP [Definition 1.1, TCC 2016], our algorithm, after a single local update satisfies $\frac{D_f^2}{2\sigma^2}$-zCDP, where we use the fact that $ \\| \operatorname{grad} f_i(z_{i,t}^r;\mathcal{B}_{i,t}^r)\\| \leq D_f$.
>
> ii) Then, according to the composition theorem [Lemma 2.3, TCC 2016], the total privacy loss after $R$ rounds of communication is $\frac{\tau R  D_f^2}{2\sigma^2}$-zCDP.
>
> iii) Finally, we convert $\rho$-zCDP to standard $(\epsilon, \delta)$-DP [Proposition 1.3, TCC 2016], which includes the quantitative relationship between the total privacy loss and the variance $\sigma^2$. Given a privacy budget, we can calculate the corresponding $\sigma^2$ to ensure that the algorithm satisfies differential privacy throughout its entire execution.
>
> The above differential privacy mechanism can defend against scenarios where the server is honest but curious, as well as where eavesdroppers can intercept the upload and download channels. Our algorithm can also be extended to other threat scenarios, with further efforts on the algorithm design and privacy analysis.
>
> [TCC 2016] Bun M, Steinke T. Concentrated differential privacy: Simplifications, extensions, and lower bounds[C]//Theory of cryptography conference. Springer Berlin Heidelberg, 2016.
>
> 6) Thank you very much for your valuable feedback, which allows us to clarify and improve our work.
> Our theoretical result, Theorem 4.3, is applicable to varying levels of data heterogeneity because our algorithm completely overcomes client drift caused by heterogeneous data and multiple local updates. First, we do not make any assumptions about the similarity of data across clients; second, we fully address client drift, as reflected in our convergence error, which eliminates the error associated with the degree of data heterogeneity. Please also refer to the response to Q4.
>
> We sincerely appreciate your thoughtful and constructive feedback! We believe our responses have thoroughly addressed all concerns, leading to significant improvements and a stronger overall paper. We would be truly grateful if you could kindly reassess our work and consider raising the score.

---

### Official Review · Reviewer_GhoW · 2024-07-12

**Soundness:** 2
**Presentation:** 1
**Contribution:** 3
**Rating:** 3
**Confidence:** 4

**Summary:**

The authors introduce a submanifold into the setting of federated learning. Utilising Riemannian gradients for optimisation, they show different properties of the proposed method, such as avoiding client drift and being computationally cheaper.

**Strengths:**

- The computational cost is advantageous, which could make new problems feasible.
- The federated learning aspects seem excellent.

**Weaknesses:**

- A central point of the paper is to not work on the manifold directly, but to utilise a projection in order to guarantee proximity. This moves the problem away from working on manifolds directly - This could be advantageous, but the presentation as working on a manifold is unnecessary and misleading.
- The presentation is unclear - while restricting themselves to only Euclidean immersions using the Euclidean metric, they state several results for which it is unclear wether these results are for general manifolds or only for this case. For the general case, the are certain problems (i.e. injectivity radius) which would need attention.

- The Lipschitz continuity in eq(3), needed for Lemma 1.2, is proven in [33] in theorem 4.8 using the Cauchy-Schwarz inequality. The generalisation to manifolds is not apparent.

[33] "Francis H Clarke, Ronald J Stern, and Peter R Wolenski. Proximal smoothness and the lower-C2 property. Journal of Convex Analysis, 2(1-2):117–144, 1995"

**Questions:**

-  Please clarify the theoretical results in terms of assumptions on the manifold needed.
-  How exactly is the proximal operator carried out? If this operator is restricted to points on the manifold, is the knowledge of which points belong to the manifold known? If yes, why not use that directly? As the proof of Lemma A.1 depends on belonging to the manifold, this appears pivotal to the concept. In [33], the proximal operator is only used in real Hilbert spaces, so belonging to a manifold is a problem arising only now.
-  Line 269 ff.: The comparison of the theoretical result with [13] seems odd, as the Li et al. do not use a proximal operator. As this operator massively eases the problem, this should at least be noted clearly - these results improve computational efficiency on another problem setting. Also, Li et al introduce a "tangent mean", where they appear to ignore the different bases of tangent spaces, more specifically that tangent spaces will differ in density in the presence of curvature. Both results are likely to not generalise to any given smooth manifold.

[13] "Jiaxiang Li and Shiqian Ma. Federated learning on Riemannian manifolds. arXiv preprint arXiv:2206.05668, 2022."

**Limitations:**

Limitations are properly addressed.

---

> ### Author Rebuttal · Authors · 2024-08-06
>
> Thank you very much for your careful review and valuable comments!
>
> Weaknesses
>
> 1)  Thank you for your comment and the opportunity to clarify this point. In our algorithm design, we not only utilize projection but also exploit the manifold geometry, specifically the Riemannian gradient, to create a more efficient algorithm. Theoretically,  we use the manifold geometry to define the optimality metric $\mathcal{G}_{\tilde{\eta}}(x)$ and establish a quasi-isometric property, i.e.,  $1/2\\|\operatorname{grad} f(x)\\|\leq\\|\mathcal{G}{\tiny \tilde{\eta}} ( x)\\|\leq 2\\|\operatorname{grad} f( x)\\|$   (see Line 259, Lemmas A.1 and A.2), which is crucial to establishing the convergence of our algorithm. It is challenging to verify this inequality using the Euclidean gradient, which ignores the manifold geometry, leading to the possibility that an algorithm that employs the Euclidean gradient instead of the Riemannian gradient may fail to converge. Therefore, both algorithm analysis and design rely on the effective utilization of manifold geometry.
>
> 2) All results are derived and proven for Euclidean immersions and the Euclidean metric. In the
> submitted version of the manuscript, we stated this on Lines 132 and 135: “Throughout the paper,
> we restrict our discussion to embedded submanifolds of the Euclidean space, where the associated
> topology coincides with the subspace topology of the Euclidean space. · · · In addition, we always
> take the Euclidean metric as the Riemannian metric.” We will revise the manuscript to ensure that
> there are no misunderstanding that our results are restricted to this case.
>
> 3) The proof of Theorem 4.8 in [33] relies on the uniqueness of the projection and the application of the Cauchy-Schwarz inequality.  As we mentioned in Lines 161-162 in the submitted version of the manuscript, "we introduce the concept of proximal smoothness that refers to a property of a closed set, including $\mathcal{M}$, where the projection becomes a singleton when the point is sufficiently close to the set." Because the projection is unique, Theorem 4.8 in [33] still holds in our case.  However,  Theorem 4.8 in [33] does not necessarily extend to general Riemannian manifolds. We will emphasize this point in the revised version.
>
> Questions
>
> 1) We assume that the manifold is a compact and smooth submanifold embedded in $\mathbb{R}^{d \times k}$, as we mentioned below Eq.(1). We will state the assumption clearly in our theorems in the revised manuscript.
>
> 2) These are good questions. We will address them one by one.
>
> i) The proximal (or projection) operator can be calculated explicitly for many common submanifolds, see [SIOPT 2012].   For the Stiefel manifold, for example, the closed-form expression for the projection operator of a matrix $X$ is $\mathcal{P}_{\mathcal{M}}(X) = X(X^\top X)^{-1/2}$; see Proposition 7 in [SIOPT 2012].
>
> [SIOPT 2012] P-A. Absil, and Jerome Malick. "Projection-like retractions on matrix manifolds." SIAM Journal on Optimization 22, no. 1 (2012): 135-158.
>
> ii) Under Assumption 2.3 and the step size condition (11), we guarantee that the projection used in our algorithm is a singleton. In our algorithm, we calculate
> $\mathcal{P}\_\mathcal{M}(\hat{z}\_{i, t+1}^r)$
> and $\mathcal{P}\_\mathcal{M}(x^r)$ which both lie on the manifold. Neither $\hat{z}\_{i, t+1}^r$ nor $x^r$ are necessary on the manifold, but we require that $\hat{z}\_{i, t+1}^r$ and $x^r$ are sufficiently close to the manifold, specifically within a tube $\overline{U}\_\mathcal{M}(\gamma):=\\{x: \operatorname{dist}(x, \mathcal{M}) \le \gamma \\}$. We ensure that this condition is met by restricting the step size to be small enough in our analysis. In our algorithm, $\hat{z}\_{i,\tau}^r$, which is the last local model during the local updates, is uploaded, and $x^r$ is downloaded. The variable $\hat{z}\_{i,\tau}^r$ is essential for the server to extract the average local gradient information by simply averaging all $\hat{z}\_{i,\tau}^r$ (since the nonlinearity of $\mathcal{P}\_\mathcal{M}(\cdot)$ prevents this from being done directly through $\mathcal{P}\_\mathcal{M}(\hat{z}_{i,\tau}^r)$). Meanwhile, $x^r$ is used locally by each client to construct the correction term $c_i^r$ without requiring additional communication.
>
> 3) The work [13] appears to be the only FL algorithm that can deal with manifold optimization problems of a similar generality as ours.
> The manifold we consider is a compact, smooth submanifold embedded in $\mathbb{R}^{d \times k}$, which is more restrictive than the manifolds discussed in [13] but it covers many common manifolds such as the Stiefel manifold, oblique manifold, and symplectic manifold.
> We will make this point noted clearly in our revised manuscript.
>
> In [13], the inverse of the exponential map is used to pull back the local updates to the same tangent space, allowing for the computation of the tangent mean.  However, the convergence in [13] is shown only for two specific settings: (1) $\tau=1$, and (2) $\tau>1$ but with only a single client participating in each training round. By leveraging the structure of the compact smooth submanifold, we demonstrate convergence for $\tau \geq 1$ with full client participation.
>
> We genuinely appreciate your thoughtful and constructive feedback. Each of your comments has been carefully considered, resulting in significant improvements and a stronger overall paper. We believe that our responses have thoroughly addressed all the concerns raised. We would be most grateful if you could kindly consider raising the score!

---

> > ### Comment · Reviewer_GhoW · 2024-08-13
> >
> > Thank you for the clarifications!
> >
> > While in general the answers improved my understanding of your point of view, I am still convinced that the presentation is misleading. The proposed method seems like a good improvement in the area of federated learning, but the writeup is currently letting the reader believe the method is working on any smooth submanifold. In the paper and the rebuttal (Weaknesses, 2.) it becomes clear that the results only hold for the choice of Euclidean space, so the presentation as manifolds is unnecessary.
> >
> > Some specific follow-ups:
> >
> > **Weaknesses**
> >
> > 1. If the setting is only Euclidean space, what is the manifold geometry you are considering?
> >
> > **Question**
> >
> > 2. i) The existence of the solution for the Stiefel manifold is obviously nice, but it is not used in the work. As it is noted by now that the given work does not generalise to manifolds, this does not appear necessary.
> >
> > I want to reiterate, I believe this is a valuable contribution to federated learning and improves on current difficulties. To reflect this, I am updating my scores. The impression of working on manifolds is massively disadvantageous for clarity of reading and precisely understanding the contribution. With a major revision changing the narrative away from manifolds I'd be happy to accept this paper.

---

> > > ### Author Response · Authors · 2024-08-13
> > >
> > > Dear Reviewer GhoW,
> > >
> > > Thank you for engaging in the discussion with us, and for your valuable feedback on our manuscript. We regret that our introduction misled you. It was never our intention, and we will certainly update it to ensure that the scope and limitations of our work are clear from the start.
> > >
> > > With this said, the manifold geometry is critical to our work, and it is not clear if and how similar results could be derived by instead focusing on general nonconvex constraints in Euclidean space.
> > >
> > >
> > > **For the weakness:**
> > > * We use the manifold geometry to compute the Riemannian gradient, i.e., ${\rm grad} f_{il}(z_{i,t}^r;\mathcal{D}\_{il})$ in our Algorithm 1. Specifically, the geometry of the manifold is essential for computing both the tangent space of the manifold and the Riemannian gradient (which is the projection of the Euclidean gradient onto the tangent space). In contrast, for a general nonconvex constraint set in the Euclidean space, determining the tangent space and its projections can be challenging. The Riemannian gradient is critical for both our algorithm design and theoretical analysis.
> > >
> > > * Furthermore, the projection operator onto the compact smooth submanifold embedded in the Euclidean space has the property of proximal smoothness. The geometry of the considered manifold helps to  establish
> > >    a quasi-isometric property between $\mathcal{G}\_{\tilde{\eta}}(x)$ and   ${\rm grad} f(x)$, i.e., $1/2 \\| {\rm grad} f(x) \\|\le \\|\mathcal{G}\_{\tilde{\eta}}(x)\\| \le 2\\| {\rm grad} f(x) \\|$ (see Line 259, Lemmas A.1 and A.2).
> > >
> > > These benefits are not always available for a nonconvex constraint set in the Euclidean space. We do not currently have a clear path for how to extend our results to general manifolds or nonconvex constraint sets, but this would be an interesting topic for future studies.
> > >
> > > **For the Question:**
> > >
> > > In our Algorithm 1, we calculate $z_{i,t}^r:= \mathcal{P}\_{\mathcal{M}}(\hat{z}\_{i, t}^r)$ and $\mathcal{P}\_{\mathcal{M}}(x^r)$,  where $z_{i,t}^r:=\mathcal{P}\_{\mathcal{M}}(\hat{z}\_{i, t}^r)$ ensures that our algorithm calculates the Riemannian gradient $\operatorname{grad} f_{il}(z_{i,t}^r; \mathcal{D}\_{il})$ at a point on the manifold, and $\mathcal{P}\_{\mathcal{M}}(x^r)$ ensures that the global model $\mathcal{P}\_{\mathcal{M}}(x^r)$ on the server satisfies the manifold constraint. In the case of the Stiefel manifold, we take advantage of the explicit expressions of the Riemannian gradient ${\rm grad} f_{il}(z) = \nabla f_{il}(z) - z \\, {\rm sym}(z^\top \nabla f_{il}(z))$ and the projection operator $\mathcal{P}\_{\mathcal{M}}(x) = x(x^\top x)^{-1/2}$ to obtain superior performance of our algorithm compared with the state-of-the-art, as shown in the numerical experiments. As stated in Eq.(1), the constraint set $\mathcal{M}$ can be any compact smooth submanifold embedded in the Euclidean space, which includes nonconvex compact sets in the Euclidean space with a smooth submanifold structure.
> > >
> > > We appreciate that you view our paper as a valuable contribution to federated learning, and we hope that our additional explanations have addressed your remaining technical concerns.
> > > We will revise our manuscript to avoid potential misunderstandings about the scope and limitations of our work, and would appreciate it if you could kindly consider raising your rating.

---

### Official Review · Reviewer_qvSZ · 2024-07-13

**Soundness:** 3
**Presentation:** 3
**Contribution:** 3
**Rating:** 7
**Confidence:** 3

**Summary:**

This article proposes a new algorithm for non-convex optimization on smooth manifolds in the federated learning setting. The algorithm is proved to have a sub-linear convergence guarantee, and numerical experiments on kPCA and low-rank matrix completion demonstrate the convergence and efficiency of the proposed algorithm.

**Strengths:**

Optimization on manifolds in the federated learning setting is a challenging topic, and this article seems to make visible contributions on this direction. The theoretical analysis also appears to be novel.

**Weaknesses:**

I do not see obvious weaknesses of the proposed algorithm, but one potential issue is how difficult the projection operator $\mathcal{P}\_{\mathcal{M}}$ is for common manifolds. Can the author(s) provide some known examples in which the expression of $\mathcal{P}_{\mathcal{M}}$ has closed forms? How is the projection operator for the Stiefel manifold computed in the numerical experiments?

**Questions:**

Besides the question given in the "Weakness" section, there is one more regarding the experiments. Although the theoretical analysis in Section 4 gives a sub-linear convergence rate, the numerical experiments mostly demonstrate a linear-like convergence pattern. Do you have any insights or conjectures for this phenomenon?

**Limitations:**

The author(s) have discussed the limitations of the proposed algorithm in Section 6.

---

> ### Author Rebuttal · Authors · 2024-08-06
>
> We would like to express our sincere gratitude for your careful review and positive assessment of our work!
>
> Weaknesses
>
> 1) The projection operator can be explicitly calculated for many common submanifolds, as discussed in [SIOPT 2012]. For the Stiefel manifold, the closed-form expression for the projection operator of a given matrix $X$ is $\mathcal{P}_{\mathcal{M}}(X) = X(X^\top X)^{-1/2}$; see Proposition 7 in [SIOPT 2012].
>
> [SIOPT 2012] P-A. Absil, and Jerome Malick. ``Projection-like retractions on matrix manifolds." SIAM Journal on Optimization 22, no. 1 (2012): 135-158.
>
> Questions
>
> 1) Thank you very much for your careful review. Although the kPCA problem is nonconvex, it has certain properties
> similar to those of a strongly convex function, e.g., the Lojasiewicz property [MP 2019]. Similar
> phenomena have also been observed in manifold optimization on low-rank matrices [NeurIPS 2021,
> AAAI 2022]. While challenging, it would be interesting to consider local regularity conditions such
> as the Lojasiewicz property and establish convergence rates faster than sublinear.
>
> [MP 2019] Huikang Liu, Anthony Man-Cho So, and Weijie Wu. “Quadratic optimization with
> orthogonality constraint: explicit Lojasiewicz exponent and linear convergence of retraction-based
> line-search and stochastic variance-reduced gradient methods.” Mathematical Programming 178,
> no. 1 (2019): 215-262.
>
> [Neurips 2021] Ye T, Du S S. Global convergence of gradient descent for asymmetric low-rank matrix
> factorization[J]. Advances in Neural Information Processing Systems, 2021, 34: 1429-1439.
>
> [AAAI 2022] Bi Y, Zhang H, Lavaei J. Local and global linear convergence of general low-rank
> matrix recovery problems[C]//Proceedings of the AAAI Conference on Artificial Intelligence. 2022,
> 36(9): 10129-10137.

---

> > ### Comment · Reviewer_qvSZ · 2024-08-12
> >
> > Thanks for the response. My questions have mostly been addressed.

---

> > > ### Author Response · Authors · 2024-08-12
> > >
> > > Thank you very much for the reply and the review!

---

### Official Review · Reviewer_q9zL · 2024-07-15

**Soundness:** 3
**Presentation:** 4
**Contribution:** 3
**Rating:** 7
**Confidence:** 3

**Summary:**

This paper addresses several computational challenges in federated learning on manifolds, by using a simple projection mapping to bypass the need for exponential mappings, logarithmic mappings, and parallel transports needed in prior work. The proposed technique also enables multiple local updates and full client participation. Theoretical iteration complexity is provided and various numerical experiments are conducted to demonstrate the sample-communication efficiency of the proposed method.

**Strengths:**

- The paper is well-written, with its novelties and contributions clearly articulated. The development flows naturally, providing sufficient intuitions and implications throughout.
- The paper identifies the computational challenges in federated learning on manifolds, specifically involving exponential mappings, logarithmic mappings, and parallel transports, and effectively addresses these challenges using a simple projection mapping.
- The idea behind the methodology is both intuitive and effective.

**Weaknesses:**

I am overall satisfied with the originality and quality of the paper, with some suggestions:

- Nonconvexity
    - First of all, the authors should define the convexity of functions on manifolds, especially since submanifolds are considered. Is the convexity defined in the ambient space or on manifolds?
    - Based on my reading, the nonconvexity of the problem involves both the nonconvexity of the manifold constraint and the nonconvexity of the objective function. As mentioned in the paper, the manifold constraint can be incorporated into the objective function as an indicator function. The authors should clearly state which aspect of nonconvexity they refer to.
    - For example, in the first sentence of Paragraph *Federated learning on manifolds*, "Since existing composite FL [...] the loss functions is convex, these methods and their analyses cannot be directly applied to FL on manifolds." It is not crystal clear whether the non-applicability is due to the nonconvexity of the manifold constraint or the objective function considered.
    - Also, it is mentioned that the algorithm is inspired by proximal FL for strongly convex problems in Euclidean spaces. Then based on my understanding, the contribution lies in not only handling the manifold constraint, but also the nonconvexity of the objective function. However, the latter part is not discussed in Section Introduction. Could the authors clarify this? The authors should also mention that related work [13] can also handle geodesically nonconvex objective functions.
- Numerical experiments
    - Most experiments use full gradients instead of stochastic gradients.
    - In the kPCA experiment, the number of clients is $1 = n = \eta _{g}^{2}$ according to Theorem 4.3. Could the authors clarify this?
- Notation and typos
    - I would suggest moving some necessary notation from Appendix A.1 to the main text to make the analysis more self-contained.
    - Is $D$ in the definition of $L_{\mathcal{P}}$ in Theorem 4.3 missing a subscript $f$?
    - The third line of Paragraph *Theoretical contributions* is missing a closing parenthesis.

**Questions:**

- The proposed methodology and underlying ideas rely heavily on the geometry of the ambient space. Could the authors comment on the difficulty of extending the methodology to more general manifolds where the ambient geometry is not available, or where the Riemannian geometry is not induced by the ambient space?
- Could you provide intuitive explanations for the step size constraints in Eq. (11)? Are these constraints purely for restricting the iterates to remain within the $2\gamma$-tube, or do they also serve to control the client drift?

**Limitations:**

Limitations and open problems are adequately discussed.

---

> ### Author Rebuttal · Authors · 2024-08-06
>
> Thank you for your careful review and positive assessment of our work!
>
> Suggestions
>
> Nonconvexity
> 1)  All the concepts related to convexity and nonconvexity are defined in the ambient space, which is the Euclidean space for the considered submanifolds. We will state this clearly in the revised manuscript.
> 2)  Yes, the nonconvexity involves both the nonconvexity of the manifold constraint and the nonconvexity of the objective function.  When the manifold constraint is incorporated into the objective function as an indicator function, the resulting objective function becomes composite, with both the smooth and nonsmooth components being nonconvex. We will state this clearly in the final version of the paper.
> 3) It is due to the nonconvexity of the manifold constraint.
> The "existing composite FL" mentioned here refers to the related work discussed in the previous paragraph, specifically Composite FL in Euclidean space ([19]-[23]). When we refer to composite FL in Euclidean space ([19]-[23]), the composite objective function consists of two parts: a smooth part and a nonsmooth part. All of these works ([19]-[23]) require the nonsmooth part to be convex. For the smooth objective function, except for [21] and the follow-up [22], which allow the smooth objective function to be nonconvex, the others require the smooth objective function to be convex or strongly convex. When we incorporate the manifold constraint as an indicator function into the objective function, it means that the nonsmooth function in the composite objective also becomes nonconvex. This results in the methods in [19]-[23] not being directly applicable. We will clarify in the revised version what specific nonconvexity we are referring to.
> 4) Thank you very much for your suggestion. We should indeed mention that the related work [13] can also handle geodesically nonconvex objective functions. Our contribution concerning the nonconvexity of the smooth objective function is to compare against existing composite FL work in Euclidean space. Among these, apart from [21] and its follow-up work [22], the other works [19], [20], and [23] all require the smooth objective function in the composite objectives to be convex or strongly convex.
>
> Numerical experiments
> 1) Thank you for your comments.
> Compared to other related algorithms, our algorithm demonstrates significant advantages in overcoming client drift, reducing communication quantity, and saving computation time. Therefore, in the comparative experiments, we have chosen to focus on these aspects. We employ full gradients to conduct ablation studies to eliminate the interference of stochastic gradients, thereby better demonstrating our algorithm's ability to handle client drift, while also showing advantages in terms of communication quantity and computation time. The impact of stochastic gradients is predictable and aligns with classical conclusions from single-machine SGD algorithms. As shown in Figure 3, the error caused by stochastic gradients can be reduced by increasing the mini-batch size.
> 2)  Thank you very much for allowing us the opportunity to clarify this point.
> To facilitate comparison with other algorithms, we fixed $\eta_g=1$ and then manually adjusted the step size $\eta$ to its optimal value. In our analysis, the condition $\eta_g = \sqrt{n}$ is used only in the last step of our analysis, when combining Lemma A.3 and (39) to derive the recursion on the Lyapunov function in (40). For all analysis results prior to inequality (40), we retained $\eta_g$ without substituting a specific value, and they still hold for $\eta_g=1$. We set $\eta_g = \sqrt{n}$ when deriving (40) because this setting conveniently cancels out the number of clients $n$, leading to a more concise result in (40). We acknowledge that the upper bound in (40) obtained with $\eta_g = \sqrt{n}$ may not be optimal. Retaining $\eta_g$ allows the server and clients to use different step sizes, increasing flexibility, which aligns with existing well-known FL literature, such as [18], [19], and [23].
>
> Notation and typos
> 1) Agree.
> 2) In the notation $D^2 \mathcal{P}_{\mathcal{M}}(x) $, $D^2$ represents the second-order derivative. We will introduce this notation more carefully in the revised manuscript to avoid confusion.
> 3) Agree.
>
> Questions
> 1) The ambient Euclidean structure of the considered submanifold is crucial in the analysis of proximal smoothness and the proposed method. Note that such manifolds cover many of the most widely used manifolds, including the Stiefel manifold, the oblique manifold, and the symplectic manifold.
> 2) For a general manifold where the ambient geometry is not Euclidean, it will be difficult to define the
> projection operator and the associated proximal smoothness. This will complicate the design and
> analysis of the algorithm. We will comment on this in
> the Conclusions and Limitations of the revised manuscript.
> 3) Thank you very much for your comments.
> In Eq.(11), we require $\tilde{\eta}:=\eta_g \eta\tau\le \min \\{ {\frac{1}{24ML}}, \frac{\gamma}{6D_f}, {\frac{1}{D_f L_{\mathcal{P}}}} \\}$. The term $\frac{\gamma}{6D_f}$ is to control the client drift, see (39); the third term $\frac{1}{D_f L_{\mathcal{P}}}$ is to derive (16), which indicates that the metric $\\| \mathcal{G}_{\tilde{\eta}}( \cdot ) \\| $ to zero implies that the first-order optimality condition is met; the first term $\frac{1}{24ML}$ is used to establish the Lyapunov function recursion (40) by combing the recursions of (17) and (39). There are some other conditions on the step sizes for restricting the iterates to remain within the $2\gamma$-tube during our analysis, such as in (17), (33), and (34), but they are implied by these three terms in Eq.(11).

---

> > ### Comment · Reviewer_q9zL · 2024-08-12
> >
> > Thank you for the detailed rebuttal. I am overall satisfied with the clarifications and the promised revisions, and I am happy to maintain my scores. Specifically, I believe that the authors' clarifications on the second point of the Numerical Experiments and the third point of the Questions section would be helpful in enhancing readers' comprehension of the paper; I recommend that these clarifications be included in the main text of the revised version.
> >
> > I do have one further question: Given the usage of the client drift correction mechanism, why is a step-size constraint still needed to control client drift?

---

> > > ### Author Response · Authors · 2024-08-12
> > >
> > > Thank you so much for your reply! We will incorporate the promised revisions and your feedback in the revised version of the paper.
> > >
> > > For your additional question on the step size, we give an explanation of why we need the step size condition in our convergence proof. The condition $\tilde{\eta} \le \frac{\gamma}{6D_f}$ comes from Eq.(39) and has two purposes: i) to control the error of SGD caused by noise and ii) to limit the client drift. To see this, note that Eq.(39) gives
> > > $$
> > > \mathbb{E}[f^{r+1} - f^{\star}] \lesssim \mathbb{E}\left[f^{r} - f^{\star} + A \frac{2L^2}{n\tau} 9\tau \\|\mathbf{\Lambda}^r-\overline{\mathbf{\Lambda}}^r\\|^2 + A \frac{2}{\tau n} \frac{\sigma^2}{b} \right],
> > > $$
> > > where $\lesssim$ means approximately smaller than or equal to, in the sense that we have ignored other terms that are not important for explaining the step size condition, and $A:= \frac{\tilde{\eta} }{2(1-\tilde{\eta}\rho)} $.  We require $A\le \tilde{\eta}$, which leads to the condition $\tilde{\eta} \le \frac{\gamma}{6D_f}$.
> > >
> > >
> > > To clarify why it is necessary to use a small step size to control the client drift, let us
> > >  assume $\sigma^2=0$. In the following, we will demonstrate that if $A$ is very large, we cannot establish the recursion of the Lyapunov function. Indeed, from Eq.(17), we know that
> > > $$
> > > \frac{1}{\tilde{\eta} n} \mathbb{E} \\| \mathbf{\Lambda}^{r+1} -\overline{\mathbf{\Lambda}}^{r+1}\\|^2 \lesssim \left( 4\eta^2 \tau L^2 9 \tau \right) \frac{1}{\tilde{\eta} n} \mathbb{E} \\| \mathbf{\Lambda}^{r} -\overline{\mathbf{\Lambda}}^{r}\\|^2,
> > > $$
> > > where we omit the other terms for simplicity.
> > > Adding the two inequalities, we get
> > > $$
> > > \mathbb{E}[f^{r+1} - f^{\star}]
> > > +\frac{1}{\tilde{\eta} n} \mathbb{E} \\| \mathbf{\Lambda}^{r+1} -\overline{\mathbf{\Lambda}}^{r+1}\\|^2
> > > \lesssim \mathbb{E}\left[f^{r} - f^{\star}\right]  + \left(\tilde{\eta} A 2L^2 9 +  4\eta^2 \tau L^2 9 \tau \right) \frac{1}{\tilde{\eta} n} \mathbb{E}\\|\mathbf{\Lambda}^r-\overline{\mathbf{\Lambda}}^r\\|^2.
> > > $$
> > >
> > > If $A$ is very large, then we cannot establish the recursion of the Lyapunov function. Instead, by substituting $A\le \tilde{\eta}$ and step size conditions (11), we have $\left(\tilde{\eta} A 2L^2 9 +  4\eta^2 \tau L^2 9 \tau \right)\le 1$.
> > > Thus, we need a small step size to control client drift.

---

> > > > ### Comment · Reviewer_q9zL · 2024-08-12
> > > >
> > > > Thank you for your quick response. The explanation is clear. If I understand correctly, there are two sources of client drift: drift from data heterogeneity, which is corrected using variance reduction, and drift from multiple local updates, which is controlled by the step-size constraint.

---

### Author Rebuttal · Authors · 2024-08-06

We sincerely appreciate the careful reviews and valuable comments from all the reviewers. We have provided point-by-point responses to each comment. In particular, for Reviewer 3dV9's Q1 regarding the comparison with related Riemannian methods and Q4 regarding the comparison with Euclidean baselines FedAvg and FedProx, our responses are as follows.

1) Reviewer 3dV9 Q1: Comparison with related Riemannian methods

i) Both [ICML 2020] and [Neurips 2021] consider the leading eigenvector problem, formulated as  $$\min_{x \in \mathbb{R}^d, \\|x\\| = 1} (-x^T A x) = \sum_{i=1}^n f_i(x), $$
where $A=\sum_{i=1}^n A_i $ and $f_i(x)=-x^T A_i x$. Let $ v_1$ be the leading eigenvector of $A$. Both papers consider the case when $A$ satisfies conditions that ensure that the problem has exactly two global minima: $v_1$ and $-v_1$. Both  [ICML 2020] and [Neurips 2021] establish linear convergence rates but rely heavily on this property of the global minima.

In contrast, we consider a more general manifold optimization problem, formulated as
$$ \min_{x \in \mathcal{M} \subset \mathbb{R}^{d\times k}} \frac{1}{n} \sum_{i=1}^n f_i(x),$$ where $x$ lies on $\mathcal{M}$ rather than on a sphere, and the objective functions $f_i$ are general nonconvex functions rather than only quadratic. Due to its generality, our analysis does not rely on any prior information about the global minima.

ii) Both [ICML 2020] and [Neurips 2021] assume that data across clients are drawn from the same distribution. In contrast, our work addresses scenarios where the data distribution varies across clients, introducing heterogeneity.

iii) Similar to us, the paper [ICML 2020] also considers using local updates to reduce communication frequency. However, their algorithm requires the transmission of both local models and local gradient information, which increases the communication overhead. Additionally, during local updates, parallel transportation is needed to translate the global gradient information to a tangent space in preparation for using the retraction operator, which increases the computational cost. In contrast, with our novel algorithm design, we only transmit local models and avoid parallel transportation.

iv) [NeurIPS 2021] uses quantization to reduce the amount of transmitted information per communication round, while our algorithm reduces communication frequency through multiple local updates. These two communication-saving strategies are distinct and lead to significant differences in the algorithms. [NeurIPS 2021] encodes $\operatorname{grad} f_i$ at each client and decodes it at the server. For decoding, previous local information must be translated via parallel transport. In contrast, our algorithm is designed to efficiently overcome client drift caused by data heterogeneity and multiple local updates while addressing more general manifold constraints. Through an innovative algorithm design, we can achieve both communication and computation efficiency.

In the revised manuscript, we will compare our method with [ICML 2020] and [Neurips 2021].

2) Reviewer 3dV9 Q4: Comparison with related  Euclidean methods

 i) For FedAvg, the paper [AAAI 2019] studied the unconstrained nonconvex problem.

 [AAAI 2019] Yu H et al. Parallel restarted SGD with faster convergence and less communication: Demystifying why model averaging works for deep learning. AAAI 2019.

Under the assumption of bounded second moments $ \mathbb{E}\_{\zeta_i \sim \mathcal{D}_i} \\| \nabla f_i(x;\zeta_i) \\|^2 \le G^2$ (Assumption 1), they established the following convergence bound (Theorem 1):
$$\mathcal{O}\left( \frac{\Delta^0}{\eta R\tau}  +  \eta^2 \tau^2 G^2 L^2 + \frac{\eta\sigma^2L}{n}\right).$$
Here, $\Delta^0$ represents the initialization error, and the other notation is consistent with the one used in our paper. The three terms in the error bound are attributed to initialization error, client drift, and stochastic variance, respectively. When data across clients are highly heterogeneous, $G$ can be very large, which causes the accuracy guarantee to deteriorate. In comparison, our algorithm, which overcomes the client drift issue, only includes the first term related to the initialization error and the third term related to the stochastic variance. Moreover, our algorithm does not require bounded second moments, i.e., that $ \mathbb{E}\_{\zeta_i \sim \mathcal{D}_i} \\| \nabla f_i(x;\zeta_i) \\|^2 \le G^2$.

ii) For FedProx, Theorem 6 in [1] proves that after $R=\mathcal{O}(\frac{\Delta^0}{\rho \epsilon})$ communication rounds where $\rho$ is some parameter given in [1, Theorem 4], it holds that $\frac{1}{R} \sum_{r=1}^R \\|\nabla f(x^r)\\|^2 \le \epsilon $.  However, the validity of this result requires the following conditions:

(1) A $B$-local dissimilarity condition must be satisfied, i.e.,$\frac{1}{n} \sum_{i=1}^n \\|\nabla f_i (x) \\|^2 \le \\| \nabla f(x) \\|^2 B^2$.
This condition implies that the data across clients are somewhat similar. Please refer to Theorem 4 and Definition 3 in [1].

(2) The local updates need to solve the subproblem inexactly until a $\gamma$-inexactness condition is met, where $\gamma$ must satisfy  $\gamma B < 1$ (Remark 5). This implies that when the degree of data heterogeneity is high and, thus, $B$ is large, $\gamma$ becomes very small, requiring a large number of local updates. This, in turn, leads FedProx to be inefficient in practice.

In summary, compared to FedAvg and FedProx, despite our objective function being generally non-convex and subject to non-convex manifold constraints, our algorithm still overcomes the issue of client drift caused by heterogeneous data and local updates. Our main analytical advantages over FedAvg and FedProx are twofold: first, we do not make any assumptions about the similarity of data across clients; second, we completely overcome client drift, as reflected in our convergence error, which eliminates the error caused by the degree of data heterogeneity.

---

### Decision · Program_Chairs · 2024-09-25

**Decision:**

Accept (poster)

**Comment:**

The reviewers were in disagreement about this work. While two reviewers were quite positive about this work, one reviewer found the applicability of the work to more general classes of manifolds limited and another reviewer (who gave a superficial review) briefly commented that experiments and discussion of previous work could be improved. After going over the paper myself I tend to also support its acceptance and think that the contribution is above bar, despite the applicability to somewhat restricted (but important) subclass of manifolds. I urge the authors to take the reviewer's comments into attention.